# Single-cell multiomics sequencing reveals the functional regulatory landscape of early embryos

Yang Wang [1,2,3,7], Peng Yuan[1,2,3,7], Zhiqiang Yan[1,2,3,7], Ming Yang[1,2,3,4,7], Ying Huo[1,2,3], Yanli Nie[1,2,3], Xiaohui Zhu[1,2,3], Jie Qiao [1,2,3,4,5,6✉] & Liying Yan [1,2,3,5,6✉]

Extensive epigenetic reprogramming occurs during preimplantation embryo development. However, it remains largely unclear how the drastic epigenetic reprogramming contributes to transcriptional regulatory network during this period. Here, we develop a single-cell multiomics sequencing technology (scNOMeRe-seq) that enables profiling of genome-wide chromatin accessibility, DNA methylation and RNA expression in the same individual cell. We apply this method to depict a single-cell multiomics map of mouse preimplantation development. We find that genome-wide DNA methylation remodeling facilitates the reconstruction of genetic lineages in early embryos. Further, we construct a zygotic genome activation (ZGA)-associated regulatory network and reveal coordination among multiple epigenetic layers, transcription factors and repeat elements that instruct proper ZGA. Cell fates associated cis-regulatory elements are activated stepwise in post-ZGA stages. Trophectoderm (TE)-specific transcription factors play dual roles in promoting the TE program while repressing the inner cell mass (ICM) program during the ICM/TE separation.

[1] Beijing Advanced Innovation Center for Genomics, Center for Reproductive Medicine, Department of Obstetrics and Gynecology, Peking University Third Hospital, Beijing, China. [2] Key Laboratory of Assisted Reproduction, Ministry of Education, Beijing, China. [3] Beijing Key Laboratory of Reproductive Endocrinology and Assisted Reproductive Technology, Beijing, China. [4] Peking-Tsinghua Center for Life Sciences, Peking University, Beijing, China. [5] National Clinical Research Center for Obstetrics and Gynecology, Beijing, China. [6] Research Units of Comprehensive Diagnosis and Treatment of Oocyte Maturation Arrest, Beijing, China. [7] These authors contributed equally: Yang Wang, Peng Yuan, Zhiqiang Yan and Ming Yang. ✉email: jie.qiao@263.net; yanliyingkind@aliyun.com

n mammals, embryo development starts from a unified zygote. Coincident with the first several zygotic cleavages, early embryos activate the zygotic genome, restore totipotency and further generate the inner cell mass (ICM) and trophectoderm (TE) during preimplantation development[1–3]. Failures in zygotic genome activation (ZGA) or ICM/TE lineage specification can cause early embryo developmental arrest and implantation failure in both mice and humans. With advances in low-input and single-cell epigenome sequencing, recent studies have revealed that extensive global epigenetic reprogramming, for instance, reprogramming of DNA methylation (Met), chromatin accessibility (Acc) and histone modifications, occurs in early embryos during this period[4–14]. However, it remains to be explored how these epigenome reconfigurations contribute to the establishment of proper regulatory networks of early embryos.

Acc is a hallmark of cis-regulatory elements (CREs), such as promoters and enhancers, that act coordinately with transcription factors (TFs) and epigenetic modifications to finely regulate the transcriptional activity of downstream genes and establish cell-type specific regulatory networks[15,16]. The currently optimized low-input open-chromatin sequencing techniques, such as the assay for transposase-accessible chromatin using sequencing (ATAC-seq) and low-input DNase I sequencing (liDNase-seq), are able to detect genome-wide dynamics of Acc in early embryos. However, the signal of the open regions reflects the average signal of the mixed sample, which may be confounded by highly heterogeneous and asynchronized blastomeres and even abnormal embryos[6,10,17,18]. Single-cell ATAC-seq detects only thousands of informative reads per cell on average, which might limit its application with the scarce resources of early embryos[19–21]. Moreover, because of the high heterogeneity of early blastomeres, a functional understanding of epigenomic changes requires knowledge of the transcriptional output from one individual cell. Recently, different single-cell epigenome sequencing methods and transcriptome sequencing methods have been combined to profile different combinations of molecular layers from the same individual cell, providing opportunities to explore the associations between different molecular layers[22–26]. However, factors compromising the quality of data from current single cell multi-omics technologies, such as poor genome coverage or low gene number detection, might constrain the precise interpretation of the associations between different molecular layers.

Here, we describe a technique called single-cell nucleosome occupancy, methylome and RNA expression sequencing (scNO-MeRe-seq) that effectively combines single-cell nucleosome occupancy and methylome sequencing (scNOMe-seq) with Multiple Annealing and dC-Tailing-based Quantitative single-cell RNA sequencing (MATQ-seq), showing improved performance for profiling of multiple molecular layers from the same individual cell[4,27,28]. We applied scNOMeRe-seq to analyze genome-wide Acc, Met and RNA expression (Expr) in mouse preimplantation embryos at single-cell resolution and to provide a comprehensive overview of the functional regulatory landscape in early embryos.

## Results

**scNOMeRe-seq profiles in mouse preimplantation embryos**. To simultaneously detect genome-wide Acc, Met and Expr in the same individual cell, we developed a single-cell multiomic sequencing method, scNOMeRe-seq, by combining scNOMe-seq and MATQ-seq (Fig. 1a). We employed this method to profile 233 single cells isolated from mouse preimplantation embryos at different stages with RNA data from 221 (94.8%) single cells and DNA data from 218 (93.4%) single cells passed our stringent criteria, showing a high success rate (Supplementary Fig. 1a–c; Supplementary Data 2). The DNA data showed that our method

could simultaneously detect over 15% genomic WCG/GCH sites (WCG 3.49 million, 15.8%; GCH 31.0 million, 15.5% on average per cell at around 3× sequencing depth) with improved capture efficiency compared with single-cell nucleosome, methylation and transcription sequencing (scNMT-seq) and single-cell chromatin overall omic-scale landscape sequencing (scCOOL-seq) (Supplementary Fig. 1e)[4,23]. Moreover, the DNA data showed a high GCH and low WCG methylation level at the previously defined DNase hypersensitive sites and open chromatin, supporting our method a valid and reproducible approach (Supplementary Fig. 1d)[6,10]. The RNA dataset showed high accuracy, high reproducibility, even coverage through genic regions, and high detection sensitivity for genes expressed at low levels (Supplementary Fig. 1f–i)[23,29]. More importantly, our RNA dataset could faithfully distinguish between ICM and TE cells in embryonic day (E) 3.5 blastocysts (Supplementary Fig. 1j–k), further confirming the high quality of our RNA data obtained from scNOMeRe-seq.

To detect the abnormal blastomeres in our early embryos, we analyzed the copy number variations (CNVs) with the RNA data and DNA data for each individual cell. Consistent results were obtained from both datasets regarding the inferred CNVs, even for the partial chromosome CNVs (Fig. 1b and Supplementary Fig. 2a). Unexpectedly, we also found several parthenogenetic (PG) embryos among our detected early embryos using single-nucleotide polymorphism (SNP)-separated allelic reads (Fig. 1c and Supplementary Fig. 2a). Then, we sought to determine whether the embryo abnormalities would cause aberrant embryo development. Notably, along the developmental trajectory inferred from the RNA dataset, most PG blastomeres showed delayed development after ZGA compared to that of normal and aneuploid blastomeres (Supplementary Fig. 2b, c). Although only 58% of ZGA genes were activated in PG blastomeres at the 2-cell stage, the PG blastomeres became more similar to the rest of the blastomeres beginning at the same stage after ZGA, indicating that the PG embryos were able to go through at least partial ZGA and develop to further stages (Supplementary Fig. 2b and d). The Met in PG blastomeres were clearly distinct from that in normal and aneuploid blastomeres, while Acc did not differ among PG, normal and aneuploid blastomeres (Supplementary Figs. 2f–g and 3a). Together, these results revealed that the aneuploid cells were able to undergo preimplantation development as well as proper epigenetic reprogramming; however, the PG cells showed delayed development and an aberrant Met pattern.

Furthermore, k-means clustering with the Acc of the transcription start site (TSS) could not distinguish the abnormal cells from the normal cells; however, the cells from the two clusters showed significant differences in global Acc levels and correlations between Acc and Expr at the TSS regions for each stage (Supplementary Fig. 3b). Notably, the cells from the cluster with the relatively higher Acc level (cluster_2) consistently showed lower correlations between Acc and Expr at the TSS regions, without differences in global Met levels, and correlations between Met and Expr were observed between the two clusters, suggesting that Acc changes in the cells of cluster_2 were irrelevant to the transcriptional regulation, which could be derived from biological differences, such as DNA duplication[30], or other unknown technical artefacts (Supplementary Fig. 3b). Furthermore, we detected the nucleosome-depleted regions (NDRs) using an aggregated Acc dataset from single cells in each cluster at each stage. Regardless of the genome coverage, cluster_1 (low Acc level and high correlation between Acc and Expr) exhibited more NDRs than cluster_2 for each stage (Supplementary Fig. 3c, d). The NDRs in cluster_1 at each stage showed greater fractions overlapping with previously defined open chromatin in early embryos than those in cluster_2 (Supplementary Fig. 3e, f)[4,6,10].

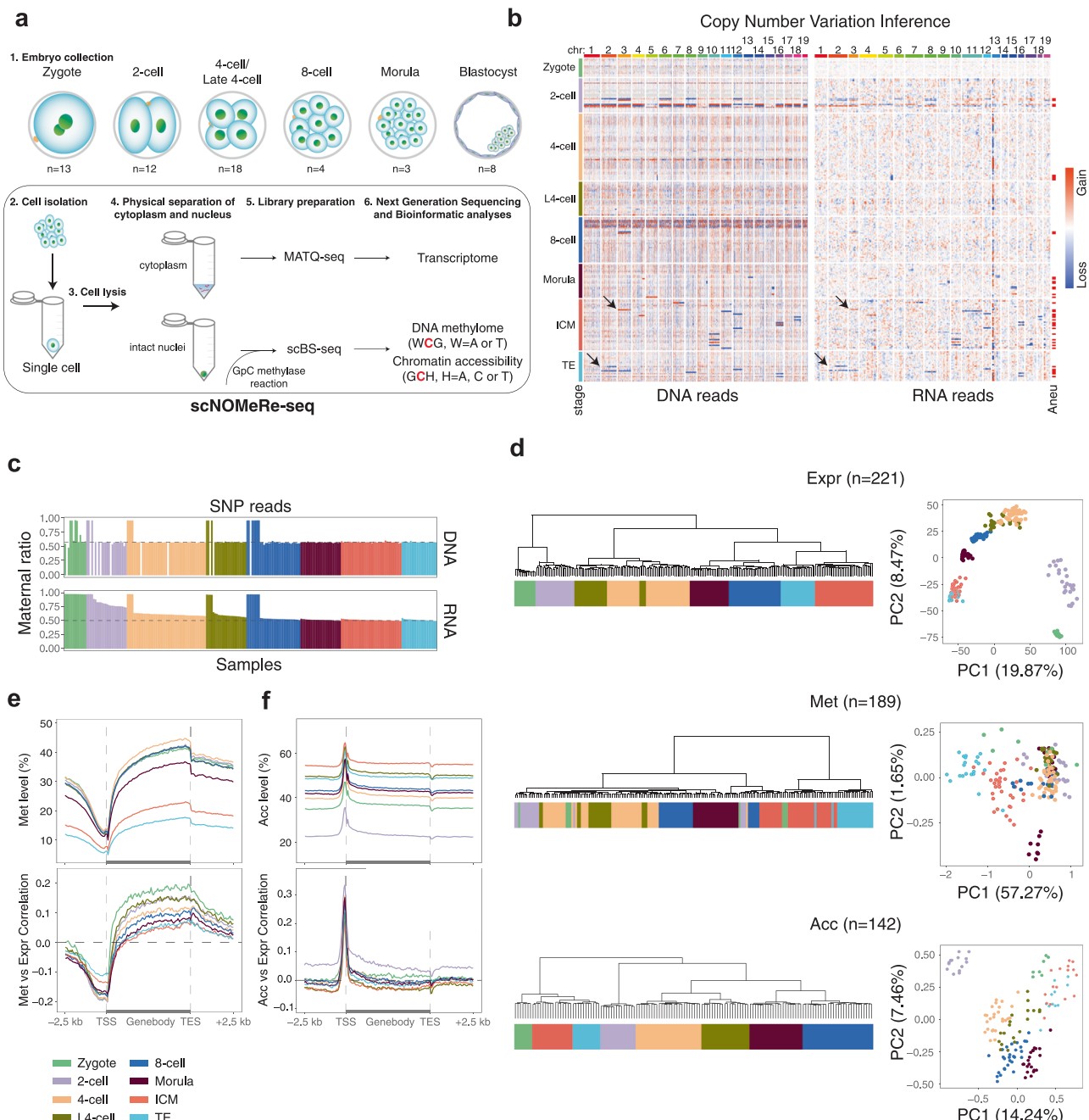

**Fig. 1 scNOMeRe-seq profiles in mouse preimplantation embryos. a** Schematic illustration of scNOMeRe-seq, including key steps, methods of library preparation, mouse preimplantation stages analyzed in this work and embryo numbers (*n*) outlined in the text. **b** Heat map showing copy number variations (CNV) inferred by DNA reads (left) and gene expression level (right, normalized each gene by average closest 100 genes) in mouse preimplantation blastomeres. Arrows indicate the examples of matched CNV inferences from DNA and RNA reads. **c** Bar plot showing the ratios of SNP tracked maternal DNA (top) or RNA (bottom) reads in total SNP tracked parental reads in each individual cell across preimplantation development. **d** Unsupervised clustering (left) and principle component analysis (right) of preimplantation blastomeres using gene expression level (top), DNA methylation level of 5 kilobases (kb) tiles (middle) and chromatin accessibility of all stage merged NDRs (bottom). n, the cell numbers of each dataset. **e** Profiles showing DNA methylation level (top), the weighted Pearson correlation coefficients of DNA methylation level vs gene expression level (bottom) along the gene bodies and 2.5 kb upstream of the transcription start sites (TSS) and 2.5 kb downstream of the transcription end sites (TES) of all genes for each stage. Met, WCG methylation; Expr, RNA expression. **f** Profiles showing chromatin accessibility (top), the weighted Pearson correlation coefficients of chromatin accessibility vs gene expression level (bottom), along the gene bodies and 2.5 kb upstream of the TSS and 2.5 kb downstream of the TES of all genes for each stage. Acc, GCH methylation. The sample size for **b** and **c** is provided in Supplementary Fig. 2a, and for **d–f** is provided in Supplementary Fig. 3g.

To focus on transcriptional regulation-related epigenome characteristics during preimplantation development, the Acc datasets of cells from cluster_2 (*n* = 68 cells) and the Met datasets of PG cells (*n* = 21 cells) were removed for downstream analysis (Supplementary Fig. 3g). Then, we explored the dynamics and associations of different molecular layers in each single cell during preimplantation development. Both unsupervised clustering and principal component analysis (PCA) revealed that cells of the same stage clustered more closely within each molecular layer (Fig. 1d), consistent with the findings of previous studies[4]. The

global Met levels were relatively stable in earlier stages but sharply decreased at the blastocyst stage (Fig. 1e, Supplementary Fig. 2e). The correlations between Met and Expr in the TSS and gene body regions showed the highest associations in zygotes and gradually decreased in the following stages (Fig. 1e). In contrast, global Acc was drastically decreased at the 2-cell stage and restored at the 4-cell stage before gradually increasing at later stages (Fig. 1f, Supplementary Fig. 2e). Notably, the correlations between Acc and Expr at the TSS regions were the most positive at the 2-cell stage among all preimplantation stages, coinciding with ZGA; this finding suggests that the drastic Acc reprogramming at the 2-cell stage might contribute to proper ZGA (Fig. 1f).

**Reconstruction of genetic lineages reveals the source of heterogeneity in early embryos.** Given the insufficient maintenance of Met levels during mitosis in early embryos, a previous study sought to reconstruct the genetic lineages of 4-cell embryos using single-cell genome-wide CpG Met datasets in both humans and mice and successfully elucidated the lineages[5]. To test whether our single-cell Met (WCG) datasets could be used to infer the genetic lineages of early embryos, we first computed the pairwise correlations among blastomeres in each individual 4-cell ($n = 10$ embryos) or late 4-cell embryo ($n = 5$ embryos) (see Methods). We repeatedly observed two pairs of cells with highly negatively correlated Met levels in each individual embryo, consistent with previous findings (Fig. 2a, b, d)[5]. Then, we validated that the two cells in each pair originated from the same mother 2-cell blastomere (Fig. 2e). We also observed a conserved pairwise correlation of Met levels among blastomeres for each analyzed 8-cell embryo ($n = 3$ embryos), implying that it might be possible to reconstruct genetic lineages for 8-cell embryos using single-cell Met datasets (Fig. 2c, d). To verify the Met correlation patterns among blastomeres at the 8-cell stage derived from the same blastomeres in 2-cell and 4-cell embryos, we microinjected FITC to label one blastomere each in 2-cell and 4-cell embryos and performed single-cell bisulfite sequencing (scBS-seq) for each individual cell when these embryos developed to the 8-cell stage. Blastomeres in 8-cell embryos derived from the same blastomeres in 4-cell embryos exhibited highly positively correlated Met levels; in contrast, 2 pairs of blastomeres in 8-cell embryos derived from the same blastomeres in 2-cell embryos exhibited highly positively correlated Met levels within pairs, but cells from different pairs exhibited highly negatively correlated Met levels (Fig. 2f, g). Therefore, these results demonstrate that we can accurately construct the full lineages from the zygote stage to the 8-cell embryo stage using single-cell Met datasets.

Furthermore, we investigated when unified zygotes generate heterogeneity in different molecular layers among blastomeres. We first computed the correlations between blastomeres within each embryo (intra-embryonic correlations) versus those between blastomeres from different embryos (inter-embryonic correlations) at the same stage for each molecular layer. We found that the intra-embryonic correlations were consistently higher than the inter-embryonic correlations for each molecular layer throughout the preimplantation development stages, suggesting highly asynchronous development among different embryos at the same stage (Fig. 2h). Moreover, the correlations in Expr levels were highest at the zygote stage and gradually decreased at later stages, suggesting that the heterogeneity among blastomeres in the same embryo was generated during ZGA and gradually increased with preimplantation development (Fig. 2h). We also noticed that the correlations in both the Met and Acc levels were highest at the 2-cell stage, indicating that the epigenome was robustly reprogrammed for each individual cell during ZGA (Fig. 2h). Leveraging this lineage tracing information, we further explored the dynamics of

heterogeneity between daughter cells during the first three cleavages. In the transcriptome, the correlations between blastomeres from the same mother cells gradually decreased during the first three cleavages, whereas the correlations between blastomeres from the same mother cells at the late 4-cell stage were comparable to those at the 4-cell stage (Fig. 2h). Moreover, the correlations between blastomeres from the same grandmother cells were higher than those of blastomeres from different grandmother cells in 8-cell embryos at the transcriptome level (Fig. 2h) (Student $t$-test, $P = 2e$-07). These results demonstrated that asymmetric cleavage might have been the major source of the transcriptome heterogeneity. We found the gradually increased transcriptome heterogeneity during the first three cleavages were highly conserved for each embryo, which enabled us to reconstruct genetic lineages of early embryos with single-cell transcriptome datasets (Supplementary Fig. 4). Although the heterogeneity in the epigenome seemed not to be associated with asymmetric cleavage, we notably observed that the correlations in Met levels between blastomeres from the same mother cells were higher in 8-cell embryos than in 4-cell and late 4-cell embryos (average correlation coefficient of 0.59 in 4-cell, 0.58 in late 4-cell and 0.74 in 8-cell; Student $t$-test, 4-cell vs 8-cell, $P = 0.018$; late 4-cell vs 8-cell, $P = 0.013$), indicating that increased Met maintenance occurs to some extent during DNA duplication at the 4-cell stage (Fig. 2h).

**Allele-specific regulation of gene expression in early embryos.** Drastic epigenetic reprogramming occurs in parental genomes after fertilization. The Acc levels of the parental genomes were comparable in most individual cells throughout preimplantation development (Fig. 3a). The Met level in the paternal allele was consistently higher than that in the maternal allele for each individual zygote (Fig. 3b). From the 2-cell to the 8-cell stage, the global differences in Met levels between parental alleles varied in the different individual cells; however, after the morula stage, the Met level in the maternal allele was consistently higher than that in the paternal allele for each individual cell (Fig. 3b). In addition, we found that 16.7%–29.7% of regions showed significant allelic differences (FDR < 0.01) in Acc levels, and 7.9%–40.2% of regions showed significant allelic differences (FDR < 0.01) in Met levels across preimplantation stages (Supplementary Fig. 5a, b). The differences in allelic Acc levels were widely distributed in the whole genome and showed no preference for a particular parental allele (Fig. 3c, Supplementary Fig. 5c). Notably, the maternal hypermethylated regions were highly enriched in genic regions, whereas the paternal hypermethylated regions were highly enriched in distal intergenic regions throughout the preimplantation stages, consistent with previous findings (Fig. 3d, Supplementary Fig. 5d)[4]. Given that the oocyte genome is highly methylated at actively transcribed genic regions and hypomethylated at intergenic regions, while the sperm genome is highly methylated at intergenic regions, our results indicate that global differences in Met levels between parental alleles in gametes could be largely maintained throughout preimplantation development[4,12].

We next sought to determine whether the allelic epigenome differences were associated with allelic transcriptional regulation. First, we overlapped the differential allelic epigenetic regions with known imprinting control regions (ICRs)[31]. Four germline ICRs overlapped with our differential allelic Acc regions, and all showed corresponding differential allelic Acc patterns in at least one preimplantation stage (Supplementary Fig. 5e). In the other hand, six known germline ICRs overlapping with differential allelic Met regions showed the expected differential allelic Met patterns throughout preimplantation development, validating the accuracy of our analysis (Supplementary Fig. 5f). Furthermore, we assessed the correlations between allelic epigenetic modification

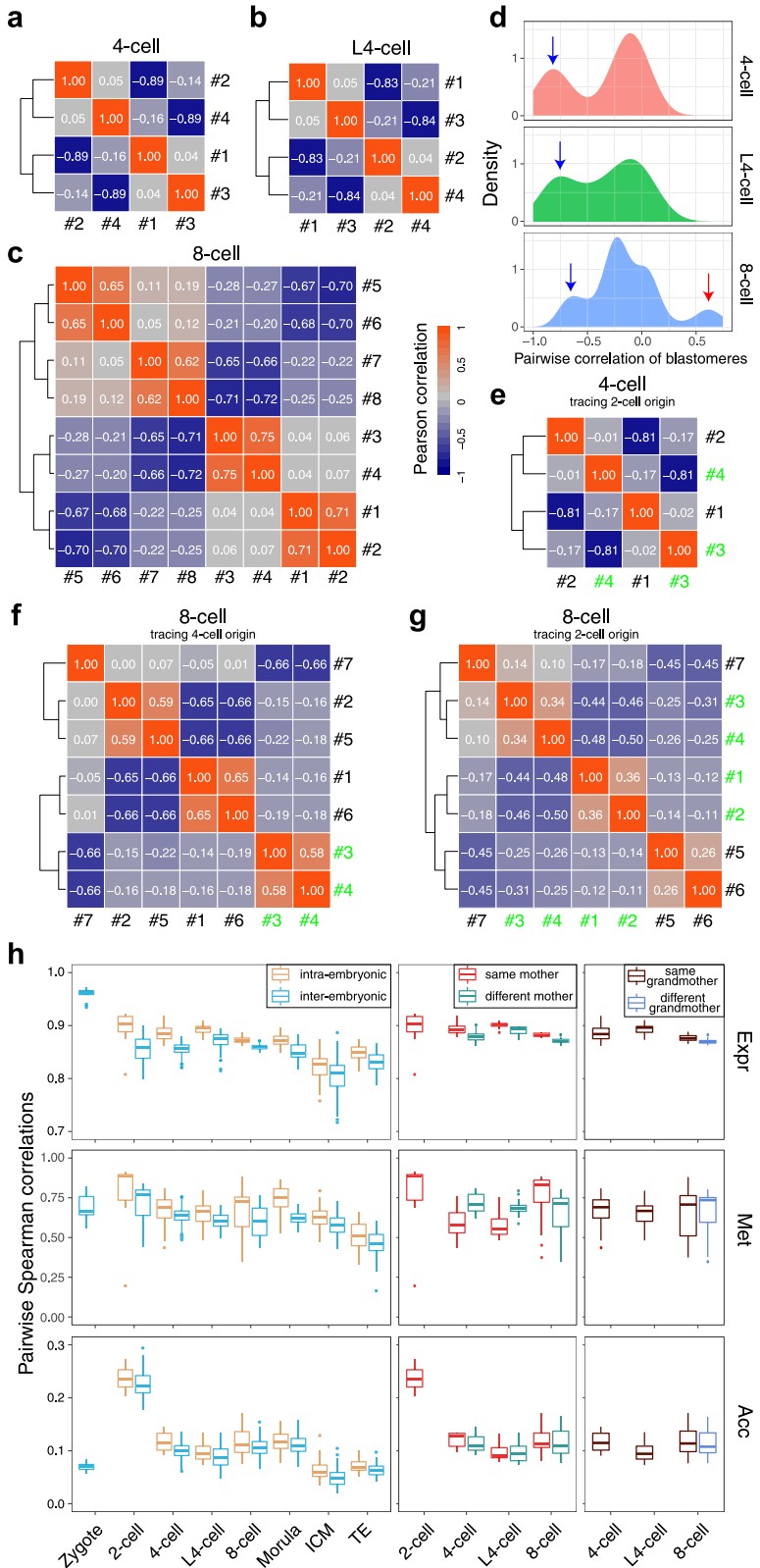

levels and Expr in each individual cell. Both parental alleles showed similar correlation patterns between allelic Acc and Expr, mimicking the overall Acc vs Expr associations (Fig. 3e). Notably, we observed clearly different correlation patterns between allelic Met and Expr in parental alleles: the Met levels of the paternal genome at the gene body regions showed no correlations with Expr, unlike those in the maternal genome (Fig. 3f). To determine

whether the allele-specific correlations between Met and Expr at gene bodies were caused by inherent correlations from maternal factors, we further compared the correlations between allelic Met and the Expr of maternal genes (transcript per million mapped reads (TPM) ≥ 1 in the zygote) with nonmaternal genes. The Met levels at gene body regions were clearly higher for maternal genes than for nonmaternal genes in maternal alleles throughout

**Fig. 2 Reconstruction of genetic lineages reveals the source of heterogeneity in early embryos. a–c** Heat map showing the Pearson correlation coefficients in representative 4-cell embryo (**a**), late 4-cell embryo (**b**), and 8-cell embryo (**c**) from the same embryo. The numbers in white color showing the Pearson correlation coefficients. **d** Distribution of the pairwise Pearson correlation coefficients for DNA methylation level of 1 Mb bins in individual blastomere from the same 4-cell embryo (*n* = 10 embryos, top), late 4-cell embryo (*n* = 5 embryos, middle) and 8-cell embryo (*n* = 3 embryos, bottom). Blue arrows indicate the pairs of blastomeres from the same 2-cell blastomere, and the red arrow indicates the pairs of blastomeres from the same 4-cell blastomere. **e–g** Heat map showing the Pearson correlation coefficients in representative 4-cell embryo (**e**) and 8-cell embryos (**f**, **g**). In **e** and **f**, #3-4 (green labeled) cells divided from the same 2-cell blastomere (**e**) and 4-cell blastomere (**f**), respectively; in **g**, #1-4 (green labeled) cells divided from the same 2-cell blastomere. The numbers in white color showing the Pearson correlation coefficients. **h** Box plot showing the pairwise Spearman correlation coefficients of RNA expression level (top), DNA methylation level of 200 kb bins (middle) and chromatin accessibility of NDRs (bottom) at indicated stages. Each box showing median levels and the first and third quartile, whiskers indicate minimum and maximum values. Points showing outliers, > 1.5 × of the interquartile range from the box. intra-embryonic, within the same embryo; inter-embryonic, in different embryos of the same stage; same mother, the blastomeres origin from a same mother blastomere; same grandmother, the blastomeres origin from a same grandmother blastomere. The sample size of each group for **h** is provided in Supplementary Fig. 3g. Source data for **h** are provided as a Source Data file.

preimplantation stages (Fig. 3g). As expected, major Met differences between parental alleles were observed in maternal genes but not in nonmaternal genes (Fig. 3g). Furthermore, we found that the correlations between maternal Met and Expr at gene body regions were clearly weaker in nonmaternal genes than in maternal genes, indicating that the observed positive correlations between maternal Met and Expr at gene body regions were mainly inherited from oocytes (Fig. 3h and Supplementary Fig. 5g).

**scNOMeRe-seq reveals a ZGA-associated regulome.** To reveal ZGA-associated CREs, we measured the correlations between the Acc of each promoter/distal NDR and the Expr of its corresponding ZGA gene (2-cell vs zygote, fold change ≥4, FDR < 0.01; Supplementary Data 3) across single cells during the transition from the zygote to the 2-cell stage. We found that 338 promoter NDRs and 7,822 distal NDRs were positively linked to 301 and 2,239 ZGA genes, respectively, while 356 promoter NDRs and 2,728 distal NDRs were negatively linked to 317 and 1,226 ZGA genes, respectively (Fig. 4a, b; Supplementary Data 4). The overall Met levels of these positively correlated CREs were lower in 2-cell embryos than in zygotes (Supplementary Fig. 6a and c). Notably, the Acc of these positively correlated CREs was specifically increased in each individual cell in 2-cell embryos, but this increase was accompanied by a drastic global Acc decrease during this period (Supplementary Fig. 6b and d). These results suggest that robust chromatin reprogramming occurs during ZGA to remove regulatory memory from gametes and rebuild the zygotic regulatory network.

To explore how ZGA is regulated in early embryos, we further comprehensively analyzed the enrichment of repeat elements and histone modifications in ZGA-associated CREs. The positively correlated CREs, but not the negatively correlated CREs, were preferentially enriched with Alu, B2, B4 and ERVL repeat classes as well as active histone modifications (H3K4me3 and H3K27ac) in both promoter and distal regions (Fig. 4c, d). H3K4me3 was gradually established at the majority of positively correlated CRE loci from the MII oocyte stage to the 2-cell stage and was colocalized with H3K27ac in both promoter and distal regions, while repressive histone modifications (H3K27me3 and H3K9me3) were gradually removed from these regions (Supplementary Fig. 7a–c). Moreover, the positively correlated CREs were clustered in regions enriched with active histone modifications and deficient in repressive histone modifications, implying a high-dimensional regulatory structure of ZGA CREs (Supplementary Fig. 7d, e). Furthermore, we investigated which TFs might be responsible for the establishment of ZGA-associated CREs. Notably, both positively correlated promoter CREs and distal CREs were highly enriched with *Arnt*, *Bcl6*, *Klf5*, *Nkx3-2*, *Nr5a2*, *Rara*, *Rarg*, *Pitx1*, and *Thrb* motifs; however, the negatively correlated CREs showed no enrichment with TFs in

either promoter or distal regions (Fig. 4e). We next calculated TF activity (see Methods) in each individual cell (Fig. 4f). Notably, we found that *Klf4*, *Nkx3-2*, *Nr5a2* and *Rarg* showed high TF activity and high expression levels in 2-cell embryos compared to zygotes (Fig. 4g). More importantly, the TF activity of *Rarg*, *Nr5a2* and *Klf4* was strongly positively correlated with the expression levels of these genes, further supporting their potential roles in regulating ZGA-associated CREs (Fig. 4h, i). It is worth noting that among these three TFs, *Klf4* already showed high expression levels and high TF activity at the zygote stage, while both *Rarg* and *Nr5a2* showed almost no TF activity at the zygote stage, implying that *Klf4*, as a maternal factor, might contribute to initiating the ZGA process as early as the zygote stage (Fig. 4f, i).

**Mutually exclusive regulome confers ICM/TE lineage segregation.** Along with gradual increases in heterogeneity among blastomeres in preimplantation embryos, establishment of cell lineage-specific transcription regulatory networks occurred beginning in unified totipotent zygotes that generated ICM and TE cells to enable further embryo development. To reveal the potential active CREs during this process, we determined the correlations between the Acc of each promoter/distal NDR and the Expr of its corresponding ICM/TE-specific expressed genes (specifically expressed in ICM: 766 genes, TE: 930 genes; Supplementary Data 5) across single cells during preimplantation development (Fig. 5a, b). The NDRs significantly correlated with ICM- or TE-specific expressed genes were termed ICM.CREs (positive: 497 in promoters, 4,086 in distal regions; negative: 210 in promoters, 1,559 in distal regions) or TE.CREs (positive: 774 in promoters, 5,109 in distal regions; negative: 424 in promoters, 3,445 in distal regions), respectively (Fig. 5a, b; Supplementary Data 6). Consistent with the ZGA-associated CREs, the positively correlated ICM/TE CREs also showed strong enrichment for active histone markers and depletion of repressive histone markers (Supplementary Fig. 8a, b). Notably, all of the known enhancers for three key ICM/TE TFs (*Pou5f1*, *Nanog*, and *Cdx2*) that we analyzed were revealed to be present in preimplantation embryos or in embryonic stem cells, confirming that the CREs identified by our correlation analysis could cover known active enhancers (Fig. 5c, d, Supplementary Fig. 8c–f)[32–34]. Specifically, we found three positively correlated CREs (#2, #3, and #4) corresponding to three known enhancers of *Pou5f1*; importantly, CRE #4 showed the highest positive correlation coefficient in our analysis, consistent with previous findings that the known enhancer corresponding to CRE #4 is the dominant enhancer regulating *Pou5f1* expression during preimplantation development (Fig. 5c, d)[32]. Thus, these results validate the accuracy of our analysis.

To explore how ICM/TE-associated regulatory networks are regulated in early embryos, we comprehensively analyzed different epigenetic molecular layers in these CREs. First, we calculated the correlations between Met and Expr for each CRE-gene pair. We observed mainly negatively correlated Met vs Expr

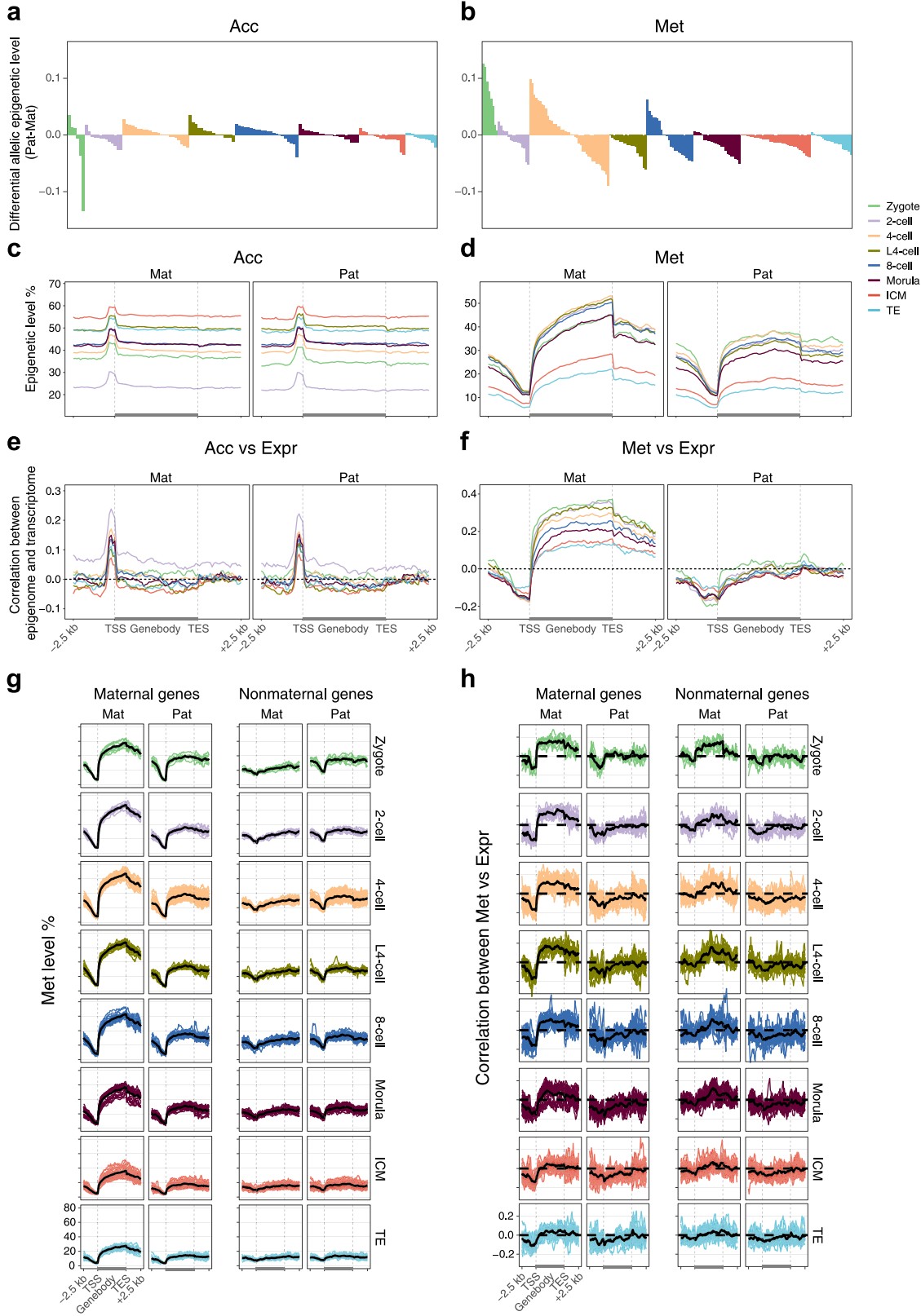

CRE-gene pairs (27 positive vs 324 negative pairs in promoters; 1,119 positive vs 4,194 negative pairs in distal regions) for the positively correlated Acc/Expr CREs, while we observed mainly positively correlated Met vs Expr CRE-gene pairs (105 positive vs 15 negative pairs in promoters; 1,504 positive vs 388 negative

pairs in distal regions) for the negatively correlated Acc/Expr CREs (Fig. 5e). These results not only reveal the complex interplay between Met and Acc in regulating ICM/TE lineage-associated regulatory networks during preimplantation development but also confirm our speculation that the CREs with Acc/

**Fig. 3 Allele-specific regulation of gene expression in early embryos. a, b** Bar plot showing differences of global chromatin accessibility (**a**) and DNA methylation level (**b**) between paternal and maternal genomes in each individual cell of preimplantation embryos. **c, d** Allelic chromatin accessibility (**c**) and DNA methylation level (**d**) around genic regions of each stage. Mat, maternal allele; Pat, paternal allele. **e, f** The Pearson correlations between chromatin accessibility (**e**) and DNA methylation level (**f**) around genic regions of maternal allele or paternal allele vs gene expression level of each stage. **g** Allelic DNA methylation level around genic regions of maternal (TPM ≥ 1 in zygote stage) and nonmaternal genes in each single cell. Solid black lines display the mean value of each stage. **h** The Pearson correlations between DNA methylation level around genic regions of maternal allele or paternal allele vs expression level of maternal and nonmaternal genes in each single cell. Solid black lines display the mean value of each stage. The sample size of each group is provided in Supplementary Fig. 3g.

Expr positive and negative correlations might be bound by activators and repressors, respectively. Regardless, the Met levels of both ICM.CREs and TE.CREs were lower in TE cells than in ICM cells, reflecting the more extensive erasure of genome-wide Met in TE cells (Fig. 5f). Next, we measured the dynamics of different molecular layers in these CREs during preimplantation development. Clearly, Acc and active histone modifications (H3K4me3 and H3K27ac) gradually increased in both positively and negatively correlated Acc/Expr CREs beginning at the 2-cell stage, while repressive epigenetic modifications (Met, H3K9me3 and H3K27me3) were already depleted in zygotes and remained depleted throughout preimplantation development in these regions, suggesting an overall priming of the epigenetic environment during ICM/TE lineage differentiation (Supplementary Fig. 9a–f). We also noticed that ICM.CREs were activated earlier than TE.CREs, as we observed clearly higher levels of active epigenetic modifications in ICM.CREs than in TE.CREs at the 2-cell stage (Supplementary Fig. 9a, b, e and 10a, b). Subsequently, TE.CREs were quickly activated and showed higher levels of active epigenetic modifications than ICM.CREs at the 8-cell stage, suggesting a stepwise activation of ICM.CREs and TE.CREs during preimplantation development (Supplementary Fig. 9a, b, e and 10a, b).

Finally, we investigated which TFs might be responsible for the establishment of differential regulatory networks in ICM and TE lineages. Notably, we found that the commonly enriched TFs in positively correlated ICM/TE CREs were ZGA drivers that showed high TF activity as early as the 2-cell embryo stage, such as *Nr5a2*, *Rarg*, *Rara*, *Bcl6*, etc., indicating that the earliest initiation of both ICM and TE programs occurs during the ZGA process (Fig. 5g and Supplementary Fig. 10c). In addition, three TFs, *Crx*, *Arnt* and *Pitx1*, were more enriched in ICM.CREs, while *Ctcf*, *Klf3/4/6/9/10*, *Gata1/2/4/6*, *Tead1/2/3/4*, *Sfpi1*, *Tcfap2a* and *Sp1/2* were more enriched in TE.CREs (Fig. 5g). We observed that TE lineage-specific TFs, such as *Tcfap2a* and *Gata* family TFs, were enriched in the negatively correlated ICM.CREs, suggesting their repressive roles in regulating the ICM program (Fig. 5g). Moreover, most TE.CRE-specifically enriched TFs showed higher activity and expression levels in TE cells than in ICM cells, while ICM.CRE-associated TFs showed higher expression levels in ICM cells than in TE cells (Fig. 5h, i and Supplementary Fig. 10c). Together, these results suggest that a mutually exclusive regulatory network is adopted to gradually establish and stabilize the different ICM and TE lineage fates, especially for the TE lineage; specific drivers of this lineage establish a TE program while repressing the ICM program, forcing the TE lineage to separate from the ICM lineage (Fig. 5j).

## Discussion

In conclusion, we have developed a single-cell multiomics sequencing technology, scNOMeRe-seq, that can be used to profile transcriptomes, DNA methylomes and chromatin accessibilities in parallel in the same individual cell with high accuracy, sensitivity and genome coverage. Taking advantage of this powerful tool, we have also characterized multiple molecular layers of

mouse preimplantation embryos at single-cell resolution and have explored the associations between different epigenome layers and transcriptional output, providing insights to enhance functional understanding of epigenetic reprogramming during mouse preimplantation development. Specifically, our results reveal that PG blastomeres show delayed development and abnormal DNA methylomes, in contrast to aneuploid blastomeres. The changes in Acc not only reflect the dynamic regulatory landscape but also may be substantially derived from asynchronous cell cycles of blastomeres that are irrelevant with transcriptional regulation, highlighting the importance of functional interpretations of epigenetic reprogramming at single-cell resolution. Using the DNA methylomes of all the individual cells within individual embryos, we reconstructed genetic lineages from zygotes to 8-cell embryos and revealed that asymmetric cleavage might be the major driver of the gradual increases in transcriptome heterogeneity among blastomeres that occur during the first three cleavages. Despite global demethylation in early embryos, allele-specific Met patterns inherited from oocytes and sperm are maintained throughout preimplantation development. The associations between Acc/Met and Expr at promoter regions in single cells are consistent with the findings from bulk samples; however, the positive correlations between Met and Expr at gene body regions are largely inherited from maternal genomes and are absent in the paternal genomes of early embryos. The Acc of parental memory-related regions appears to be substantially erased during the ZGA process and reconfigured in concert with the influences of histone modifications, Met, repeats, TFs, and possible high-dimensional chromatin structures to ensure proper activation of the zygotic genome (Fig. 4j). The overall-primed ICM/TE lineage-associated CREs are partially activated as early as the 2-cell stage and are asynchronously activated in the following preimplantation stages. Intriguingly, TE lineage-specific TFs seem to play dual roles in activating the TE program and repressing the ICM program, thereby segregating the TE fate from the ICM fate (Fig. 5j). Taken together, our findings not only provide insights into the functional regulatory landscape in preimplantation development but also elucidate the fundamental mechanisms of epigenetic regulation.

## Methods

**Embryo collection**. All animal-related experimental procedures were carried out under ethical guidelines set forth by the Animal Care and Use Committee of Peking University Health Science Center (no. LA2018261). The female mice used in this study were 6- to 8-week-old B6D2F1/J (BDF1) mice, and the male mice were 12-week-old 129S1 mice. All mice were in good health. For preimplantation embryo collection, female mice were superovulated by injection of 7.5 IU of pregnant mare serum gonadotropin (PMSG) followed by 7.5 IU human chorionic gonadotropin (hCG) 42–48 h later. Immediately after hCG injection, the female mice were placed in a cage with males. After vaginal plugs appeared, zygotes were collected from the mouse oviducts, and the embryos were further cultured in KSOM medium (Zenith Biotech). Embryos at different stages were collected 24–26 h (zygotes), 36–40 h (2-cell embryos), 52–54 h (4-cell embryos), 58–60 h (late 4-cell embryos), 66–70 h (8-cell embryos), 78–82 h (morulae) and 94–98 h (blastocysts) after hCG injection. The embryos were exposed to acidic solution (0.5% concentrated HCl) to remove the zona pellucidae and were then washed with 0.5% HSA/DPBS (Vitrolife, Gibco) to remove polar bodies and somatic cells. The embryos were further dissociated into single cells by incubating them in a trypsin

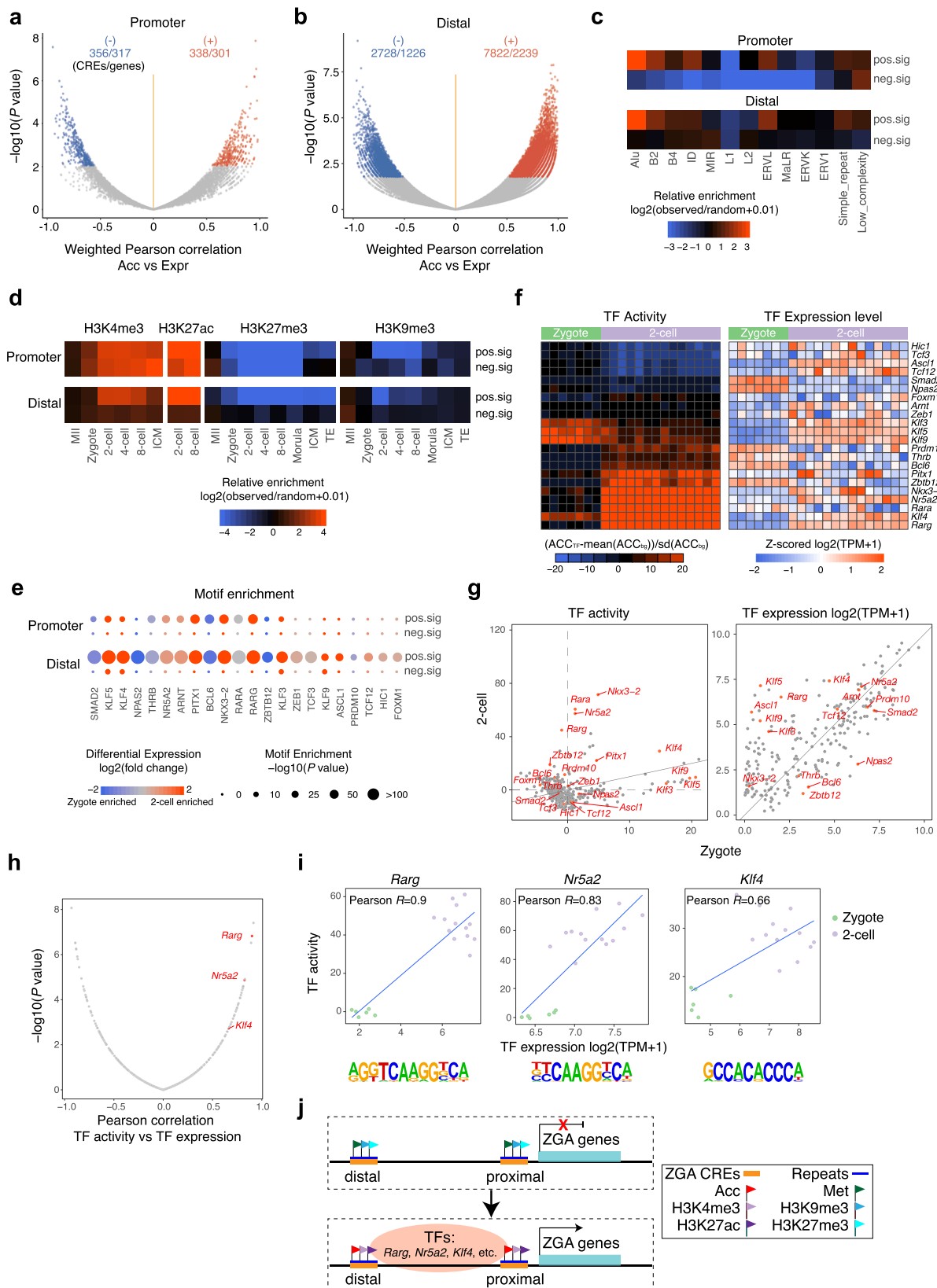

(Sigma-Aldrich) and Accutase (Millipore) solution (at a 1:1 volume ratio) for 15–50 min. Then, the embryos were washed with 0.5% HSA/DPBS three times.

**Lineage tracing with FITC injection**. One blastomere of each 2-cell stage mouse embryo ($n = 4$ embryos) was microinjected with FITC-coupled dextran (Sigma-Aldrich) to indicate its daughter cells at the 4-cell stage ($n = 3$ embryos) or its

granddaughter cells at the 8-cell stage ($n = 1$ embryo). To indicate which blastomeres of 8-cell embryos were from the same mother cells of 4-cell embryos, one blastomere of the 4-cell stage embryo ($n = 1$ embryo) was microinjected with FITC-coupled dextran. After those embryos developed to the desired stage, each embryo was dissociated into single cells as mentioned above. The FITC-labeled cells were identified under a fluorescence microscope before library preparation. All available single cells were processed for scBS-seq[35].

**Fig. 4 scNOMeRe-seq reveals a ZGA associated regulome. a, b** Volcano plot showing the weighted Pearson correlations between chromatin accessibility of promoter-NDRs (**a**) / distal-NDRs (**b**) and the expression level of corresponding ZGA genes across cells from zygote to 2-cell stage. Significant associations (FDR < 0.1) are in red (positive) and blue (negative). The number of CREs and unduplicated genes are labeled. **c, d** Heat map showing the enrichment of ZGA associated CREs in repeats (**c**) and histone modifications (**d**) (refs. [7-9]) of early embryos. **e** TF motifs identified from ZGA associated CREs. Only TFs with the $P$-value $< 1 \times 10^{-10}$ and TPM $\geq 5$ at least at one stage were included. $P$-value, binomial test in HOMER. **f** Heat map showing the TF activity (left) and expression level (right) of ZGA enriched TFs in each individual cell of zygote and 2-cell embryos. **g** Scatter plot showing the difference of TF activity (left) and expression level (right) between zygote and 2-cell stage. The genes labeled in red indicate the TFs showing significantly differential activity (left, FDR < 0.1) or expression level (right, FDR < 0.01). **h** Volcano plot showing the Pearson correlations between TF activity and expression level across cells from zygote to 2-cell stage. Significantly correlated (FDR < 0.1) TFs that enriched in ZGA associated CREs and showed higher TF activities and expression level in 2-cell embryos are labeled. **i** Scatter plot showing TF activity and expression level of *Rarg*, *Nr5a2* and *Klf4* in each single cell of zygote and 2-cell embryos. The corresponding enriched DNA-binding motifs are shown at the bottom of each TF. **j** A model showing ZGA process regulated by multiple epigenetic layers, transcription factors and repeat elements. The sample size of each group is provided in Supplementary Fig. 3g. Source data for **e** are provided as a Source Data file.

**Library preparation and sequencing**. Single cells were picked by mouth pipetting, and each was transferred into a 200 μl PCR tube containing 3 μl of cell lysis buffer (1x RT buffer (Invitrogen), 0.5% NP40, 5 mM DTT, 2 U RNase OUT (Invitrogen), and 0.2 μl of magnetic beads (Invitrogen, cat. # 65001)). The cells were lysed on ice for 10 min and vortexed for 30 s. Then, the tubes were placed on a magnet for 5 min to separate the beads (containing the intact nuclei) and supernatant (containing the RNA transcripts). The supernatant was transferred to a new PCR tube, and RNA libraries were prepared following the MATQ-seq protocol[28]. External RNA Controls Consortium (ERCC) spike-ins (Ambion) were added to the supernatant at dilutions of 1:10000-100000. For the split RNA libraries, the supernatant was split into 2 aliquots before first-strand synthesis, and the libraries were prepared separately. In parallel, the precipitated beads (containing the intact nuclei) were resuspended in 5 μl of GpC methylase reaction buffer (1× M.CviPI reaction buffer (NEB), 5 U M.CviPI (NEB), 160 μM S-adenosylmethionine (NEB), 0.25 mM EDTA (Thermo), 0.25 mM phenylmethylsulfonyl fluoride (Sigma-Aldrich), and 1 pg of lambda DNA (NEB)) and incubated in a thermocycler at 37 °C for 60 min to methylate GpC before heat inactivation for 25 min at 65 °C. After in vitro GpC methylation, 0.5 μl of protease (Qiagen, 20 mg/ml) and 10 ng of carrier RNA (Qiagen) were added into the mixture, which was incubated at 50 °C for 3 h in a thermocycler to release genomic DNA before heat inactivation at 75 °C for 30 min. Then, the genomic DNA was bisulfite-converted (Thermo), and the scBS-seq protocol was followed to prepare DNA libraries[35]. For the FITC-tracing experiments, the dissociated single cells were transferred into a 200 μl PCR tube containing 5 μl of RLT plus (Qiagen) to release genomic DNA, and then the scBS-seq protocol was followed to prepare DNA libraries. All primers used in this study can be found in Supplementary Data 1. All the libraries in this study were sequenced on an Illumina HiSeq X Ten platform with 150 bp paired-end reads. With scNOMeRe-seq, we detected 3.49 million unique WCG sites, 31.04 million unique GCH sites and the expression of 14416 (TPM > 0) GENCODE genes per cell on average.

**scRNA-seq data processing**. The raw reads were trimmed with Trim Galore (v0.4.4) to remove the primer sequences and low-quality bases (parameters: trim_galore --paired --quality 20 --phred33 --stringency 3 --gzip --length 36). The trimmed reads were aligned to the GENCODE NCBIM37 reference genome (corresponding to the University of California, Santa Cruz (UCSC) mm9 genome) with STAR software[36] with the default settings, and only the unique mapped reads with MAPQ values ≥ 50 were retained for further analysis. The reads mapped to rRNA were filtered out with RSeQC[37]. The coverages of the transcripts were evaluated with RSeQC. The reads mapped to each gene were counted with featureCounts[38] (parameters: featureCounts -p -t exon -g gene_id), and the gene expression levels were calculated using TPM values. Samples were discarded for subsequent Expr analysis that had (1) less than 9500 genes detected, (2) the library size less than 0.1 million counts, (3) reads for mitochondrial genes accounted for over 30%, or (4) reads with a fraction of the top 50 features over 40%. Then, the split cell libraries that passed quality control were merged using the DESeq2[39] R package. In total, 221 single-cell RNA libraries were retained in this study.

**Principal component analysis and hierarchical clustering of individual cells using RNA expression data**. PCA and hierarchical clustering were performed to analyze cell populations with the Expr data. Genes expressed in fewer than 6 cells were discarded. PCA was performed with the expression matrix of high variable genes (coefficient of variation ≥ 1) using the *pcaMethods*[40] R package. The *hclust* function with the *ward.D2* method was used for unsupervised hierarchical clustering.

**SC3 clustering**. To assign single cells from the blastocyst stage into ICMs and TEs and to identify markers of the two types of cells, we used the *SC3*[41] R package on the scRNA-seq data. The quality-controlled expression matrix was passed into *SC3*,

and the clusters were plotted with the sc3_plot_consensus function. The marker genes were further identified with the sc3_plot_markers function.

**Preimplantation embryo developmental pseudotime analysis**. Although PCA clustered the individual cells into their biological developmental stages, the cells within each group were not ordered by their developmental time points. Therefore, individual cells were further ordered with the *destiny*[42] R package using the top 2000 variable genes in the individual cells.

**Differential RNA expression**. Differential Expr between groups and stages was analyzed using the R package *DESeq2*.

**Copy number variation estimation using scRNA-seq data**. To evaluate the CNV effect on Expr, we inferred the CNV status of each cell by averaging the expression of the ordered genes across the genome[43]. In brief, genes were ordered along the genome by their genomic locations, and the expression of each gene was adjusted to the average expression of the 50 upstream and downstream genes. The adjusted expression values were further centered to infer the CNV status.

**scNOMe-seq and scBS-seq data processing**. The paired-end FASTQ reads were processed as two single-end FASTQ files because chimeric reads were produced during library preparation[44]. The raw paired-end FASTQ reads were trimmed to remove the first 11 bp of the random primer sequences, Illumina adapter sequences and low-quality bases with Trim Galore in single-end mode (parameters: trim_galore --clip_R1 11 --quality 20 --stringency 3 --length 30). The trimmed reads were aligned to the lambda and UCSC mm9 genome with Bismark[45] in single-end mode (parameters: bismark --bowtie2 --non_directional). The single-end-mode mapped reads were merged, and PCR duplicates were removed (PICARD) for downstream analysis. The bismark_methylation_extractor function was used to call the methylation value for each cytosine site in the genome (parameters: bismark_methylation_extractor -s --multicore 4 --gzip --cytosine_report --CX), which required at least 1x coverage at the cytosine sites. WCG (W includes A and T) and GCH (H includes A, C, and T) Met levels were calculated to represent the Met levels and Acc levels, respectively. The samples with less than 0.5 million covered WCG and 5 million covered GCH sites were discarded before downstream analysis. In total, 218 scNOMe-seq and 26 scBS-seq libraries passed the quality control steps.

**Quantification of DNA methylation and accessibility**. The Met and Acc levels were calculated as the sum of methylated reads (C) divided by the total covered reads (sum of the methylated reads and unmethylated reads (T)) for each WCG and GCH site, respectively. The Met and Acc levels of the genomic regions and single cells were measured as the average WCG and GCH levels, respectively. The TSS Met level was measured as the average WCG level within 1 kb upstream and 0.5 kb downstream of the TSS. TSS Acc was measured as the average GCH level within 200 bp upstream and 100 bp downstream of the TSS.

**Nucleosome-depleted region identification**. To identify NDRs[4], the GCH data of each single cell from the same developmental stage were first aggregated. The number of C and T reads in the GCH context within a 120 bp window with 20 bp spacing was calculated, and the significance of the difference from the genomic background was analyzed with the Chi-square test. Regions with significantly elevated GCH methylation with $P$-values $\leq 10^{-15}$, lengths ≥ 140 bp and covered GCH sites ≥ 5 were defined as NDRs. The NDRs that overlapped with the ENCODE blacklist (mm9, http://mitra.stanford.edu/kundaje/akundaje/release/blacklists/) were removed before downstream analysis. The defined NDRs were further classified as Promoter_NDRs (within 2 kb of the TSS; NDRs that overlapped with the TSS termed TSS_NDRs, while those that did not overlap with the TSS were termed Proximal_NDRs) and Distal_NDRs (at least 2 kb away from the TSS) in downstream analysis.

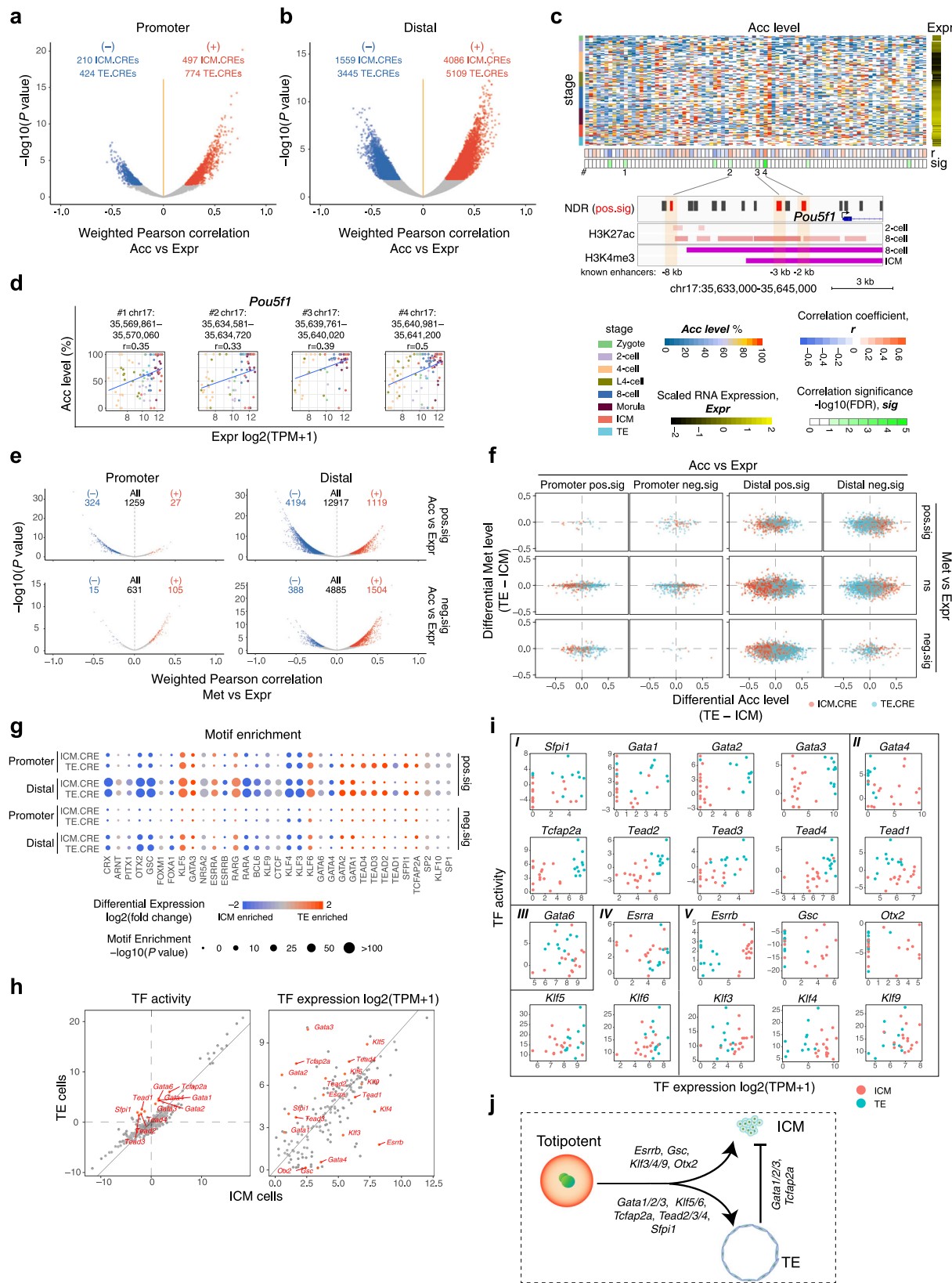

**Copy number variation detection using scBS-seq data**. To detect the CNV in each cell based on the scBS-seq data, we inferred the CNVs using the HMMcopy[46] R package. First, the sequenced reads were counted with readCounter in 1 Mb bins across the mouse genome, and the read count of each bin was adjusted by correcting for the GC content and genomic mappability. Then, the median adjusted read counts of the consecutive bins were used to infer the CNVs of the genomic regions.

Median values of more than 1.36 and less than 0.67 observed in >10 consecutive bins were defined as gains and losses, respectively.

**Principal component analysis and hierarchical clustering of blastomere DNA methylation and chromatin accessibility**. To analyze cell populations with the

**Fig. 5 Mutually exclusive regulome confers ICM/TE lineage segregation. a**, **b** Volcano plot showing the weighted Pearson correlations between Acc of NDRs and Expr of corresponding ICM/TE specific expressed genes across preimplantation development. Significant associations (FDR < 0.1) are in red (positive) and blue (negative). The number of CREs are labeled. **c** (top) Heat map showing Acc of *Pou5f1* surrounding NDRs, and Expr of *Pou5f1* in early embryos; #, the labels of positive-correlated CREs. (bottom) IGV snapshot showing the distribution of NDRs, H3K27ac and H3K4me3 peaks. Three known enhancers of *Pou5f1* are shaded. **d** Scatter plot showing the Expr of *Pou5f1* and Acc of CREs labeled in **c** in early embryos. The genomic coordinates of CREs and the correlation coefficients are shown. **e** Volcano plot showing the weighted Pearson correlation between Met and Expr of positive-correlated (top) and negative-correlated (bottom) CREs. **f** Scatter plot showing the differential Acc and Met of ICM.CREs and TE.CREs between ICM and TE cells. **g** TF motifs identified from ICM.CREs and TE.CREs. Only TFs with the P value < 1 ×10$^{-10}$ and TPM ≥ 5 at least at one stage were included. P value, binomial test in HOMER. **h** Scatter plot showing the difference of TF activity (left) and Expr (right) between ICM and TE. The genes labeled in red showing significantly differential activity (left, FDR < 0.1) or Expr (right, FDR < 0.01). **i** Scatter plot showing the TF activity and Expr in each individual cell of ICM and TE. *I*, the TFs showing higher activity and Expr in TE cells; *II*, the TFs showing higher activity in TE cells but showing higher Expr in ICM cells; *III*, the TF showing higher activity in TE cells but showing no difference in gene expression level; *IV* and *V*, the TFs showing no difference in activity but showing higher Expr in TE cells (*IV*) or in ICM cells (*V*). **j** A model showing potential TFs in driving ICM and TE lineage fates. The sample size of each group is provided in Supplementary Fig. 3g. Source data for **g** are provided as a Source Data file.

Met and Acc datasets, we performed PCA and hierarchical clustering using the WCG levels of genome-wide 5 kb tiles and the GCH levels of merged NDRs. Genomic regions that covered fewer than 3 sites were discarded. Spearman correlation coefficients were calculated with the parameter *pairwise.complete.obs*. PCA was performed using the *pcaMethods* R package. The *hclust* function with the *ward.D2* method was used for unsupervised hierarchical clustering.

**Profiling of DNA methylation and chromatin accessibility.** To profile the Met and Acc levels around genic regions (from 2.5 kb upstream of the TSS through the gene body to 2.5 kb downstream of the transcription end site (TES)), the average GCH levels and WCG levels within predefined running windows were computed for each gene in each single cell[23]. The running window was defined as a 150 bp window with a 50 bp step for the TSS upstream and the TES downstream. The gene body (from the TSS to TES) of each gene was divided into 100 equal fractions, and the running window was defined as a 2-fraction window with a 1-fraction step. To profile the Met and Acc levels around NDRs (from 3 kb upstream of each NDR to 3 kb downstream), the average GCH levels and WCG levels within a 150 bp running window with a 50 bp step were computed for each NDR in each single cell. The values of all the same genomic locations from individual cells were combined to plot the average profile for single cells. The values from cells at the same stage or in the same group were combined to plot the average profile for the stage or group, respectively.

**Correlation analysis.** To profile the relationship between Met/Acc and Expr around the genic regions, the Pearson correlation was calculated between the average WCG/GCH level within the running window (described above) and the Expr of its corresponding gene across different genes in a single cell. To calculate the associations between Met/Acc and Expr at specific genomic regions (the TSS and gene body), the Pearson correlation was calculated between the average WCG/GCH level within the region and the Expr across different genes in a single cell.

To infer the functional CREs with associated genes, we computed the correlation between the dynamic Met/Acc level of each NDR and the expression of its corresponding gene across cells during preimplantation development. All possible relationships between NDRs and genes within 100 kb of the gene (upstream of the TSS and downstream of the TES) were considered. NDRs with a coverage of fewer than 3 sites in a single cell were discarded. NDRs covered in less than 25% of the cells and nonvariable NDRs were discarded. Genes expressed at low levels (expressed (TPM > 0) in fewer than 5 cells) and nonvariable genes were discarded. We calculated a weighted Pearson correlation coefficient (using the unique WCG or GCH sites covered within the NDRs in each single cell as a weight) and tested the significance of the coefficients with two-tailed Student's *t*-tests. The P-values were further adjusted by the Benjamini-Hochberg approach.

**Allele-specific analysis of RNA expression, DNA methylation and accessibility.** The embryos in this study are from 129S1 (paternal) mice × B6D2F1/J (F1 of C57BL6NJ × DBA2J, maternal) mice. Thus, these embryos should have backgrounds of 129S1 with mixed C57BL6NJ and DBA2J. The pipeline used to determine the parental origin assignment of sequencing data from the hybrid embryos was constructed as reported, which based on traceable hybrid SNP information[47]. Specifically, we downloaded the SNPs of 129S1 (paternal in this study), C57BL6NJ and DBA2J (C57BL6NJ × DBA2J, maternal in this study) from the website of Mouse Genome Project (ftp://ftp-mouse.sanger.ac.uk/REL-1211-SNPs_Indels/). Only the informative SNPs could distinguish the paternal (129S1) and maternal (C57BL6NJ × DBA2J) genome (homozygous in parental alleles and paternal is different with maternal) were used in our analysis. For each mapped read covered the informative SNP site, the read was parsed according to the specific base at the SNP position, if the base matched the paternal allele, the read was assigned to paternal origin; if the base matched the maternal allele, the read was assigned to maternal origin. For RNA-seq data, 172,319 SNPs within the exon

regions were used to split the RNA mapped reads. The splitted allelic reads were further used to calculate the paternal and maternal expression level of each gene. For DNA data, 896,161 SNPs (SNP sites with C or T were discarded) in the whole genome were used to split the mapped reads to paternal and maternal origin. The splitted reads were further processed to calculate the paternal and maternal DNA methylation and chromatin accessibility level.

To profile the allelic Met and Acc levels around genic regions (from 2.5 kb upstream of the TSS through the gene body to 2.5 kb downstream of the TES), the average allelic WCG levels and GCH levels within predefined running windows (500 bp window with a 100 bp step for the TSS upstream and the TES downstream; 10-fraction window with a 2-fraction step for the gene body (100 equal fractions)) were computed for each gene in each single cell. Local correlations were calculated between the average allelic WCG/GCH level within the running window and the total Expr of its corresponding gene across different genes in a single cell.

The epigenetic modification levels of 500 bp tiles were extracted from the parental alleles of each single cell separately. The tiles retained in each stage were required to cover at least 3 cells per allele. The numbers of methylated sites (C) and unmethylated sites (T) were summed in each 500 bp tile per allele for each stage. Fisher's exact test was used to examine the differences between two alleles for each tile. Tiles with FDR values less than 0.01 were considered to differ significantly between alleles.

**Integrative analysis of public data and resources.** Repetitive elements (such as LINEs, SINEs, Alu elements, etc.) and related genomic annotations (such as the TSS, TES, gene body, etc.) were downloaded from the UCSC. Promoters were defined as the regions from −1.5 kb to +0.5 kb relative to the TSS. Histone modification data for mouse preimplantation embryos were downloaded from the Gene Expression Omnibus (GEO) database (GSE71434 for H3K4me3 modifications; GSE97778 for H3K9me3 modifications; GSE72784 for H3K27ac modifications; GSE73952 for H3K27me3 modifications) and integrated into our analysis. The peaks were downloaded from the GEO database. The enrichment analysis was calculated as log2 ratio for the number of observed CREs that overlap with repeats or histone modification peaks divided by the number of random regions that overlap with repeats or histone modification peaks. The signal intensity of each histone modification was calculated with an identical procedure. In detail, the downloaded raw FASTQ reads were trimmed with Trim Galore (parameters: trim_galore --paired --quality 20 --phred33 --stringency 3 --length 36). The processed clean reads were mapped to the UCSC mm9 genome using BWA with the default settings[48]. The duplicated reads were removed with PICARD software, and only unique mapped reads with MAPQ values ≥ 30 were retained. The sequencing coverage was further normalized, and the histone modification signal intensity of the investigated regions was calculated with the *Deeptools* package[49].

**Transcription factor analysis.** To find the enriched TF motifs in different CRE data sets, findMotifsGenome.pl in HOMER[50] was used (parameters: -size given -len 8,10,12). Only TFs with the P value < 1 ×10$^{-10}$ and TPM ≥ 5 at least at one stage were included.

To further infer the TF activity in each single cell, we calculated background-corrected z-scores for the average Acc of the TF-binding sites (TFBSs) for each TF in each single cell. A similar strategy has been developed for single-cell ATAC-seq[51]. More precisely, all stage-defined distal NDRs were scanned to find TFBSs using FIMO[52]. Position frequency matrices were converted from 413 Homer motifs. We kept the motif TFBSs with P-values less than 1 ×10$^{-4}$. The overlapping motifs for each TF were merged. The average Acc of the TFBSs (extended from the center to +− 50 bp) in each single cell for each TF was calculated as the raw TF activity (ACC$_{TF}$). Then, we permuted the data 1,000 times for the TFBSs for each TF in each single cell and calculated the average Acc (ACC$_{bg}$) for each permutation. For each TF in each cell, TF activity was calculated as (ACC$_{TF}$-mean (ACC$_{bg}$))/sd(ACC$_{bg}$).

**Reporting summary**. Further information on experimental design is available in the Nature Research Reporting Summary linked to this paper.

## Data availability

All sequencing data in the current study have been deposited in the GEO with the accession number GSE136718. Publicly available datasets were downloaded from the GEO database GSE71434 for H3K4me3 modifications; GSE97778 for H3K9me3 modifications; GSE72784 for H3K27ac modifications and GSE73952 for H3K27me3 modifications). All other relevant data supporting the key findings of this study are available within the article and its Supplementary Information files or from the corresponding author upon reasonable request. A reporting summary for this Article is available as a Supplementary Information file. Source data are provided with this paper.

## Code availability

Custom scripts used in this study can be downloaded from https://github.com/yang2sc/scNOMeRe-seq.

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

## Acknowledgements

This work was supported by the grants from National Key Research and Development Program (2018YFC1004000, 2019YFA0801400 and 2017YFA0105001), National Natural Science Foundation of China (31571544, 81730038, 81521002 and 31701261) and the Fundamental Research Funds for the Central Universities-Peking University Clinical Scientist Program. Y.W. was supported by Postdoctoral Fellowship of Peking-Tsinghua Center for Life Science and the grant from China Postdoctoral Science Foundation (2016M600873).

## Author contributions

Y.W. conceived the method. L.Y. and J.Q. conceived the project. Y.W., P.Y. and M.Y. performed experiments with the help of Y.H., Y.N., X.Z. Y.W., Z.Y. and Y.P. performed computational analysis. Y.W., P.Y. and Z.Y. wrote the manuscript with feedback from all authors.

## Competing interests

The authors declare no competing interests.
