## [Peer Review File · Nature Communications]

Reviewers' comments:

Reviewer #1 (Remarks to the Author):

The present manuscript by Wang et al. developed a single-cell multiomics sequencing technology (scNOMeRe-seq) that enables profiling of genome-wide chromatin accessibility, DNA methylation and RNA expression in the same individual cell. By applying this method, the authors studied the global dynamics of different molecular layers and their associations in mouse preimplantation embryos. Based on a comprehensive amount of data and detailed analysis, this paper deciphers a single-cell triple-omics map of chromatin accessibility, DNA methylation and RNA expression during mouse preimplantation development and provides many specific and novel insights into the understanding of epigenetic regulation in early embryos. The dataset would be of interest to others in the community, whereas, its weak points are the lack of finding the specific TFs/CREs/Met and the largely confirmatory nature of the results. It will substantially improve the manuscript if more analysis results of the datasets and validation work can be presented. Below please find my specific points that I feel should be addressed prior to publication.

1. This manuscript reported the method "scNOMeRe-seq" which combined scNOME-seq and MATQ-seq, the authors should demonstrate the strength of this method compared with other similar method e.g., scNMT-seq (doi: 10.1038/s41467-018-03149-4). Besides, more evaluation and comparison including chromatin accessibility and methylation should be presented to support this method a valid and reproducible approach. For example, methylation coverage, whether the accessibility data affected by endogenous GpC methylation, Met and Acc profiles at DNase hypersensitive sites etc.
2. Extended Data Fig. 1d showed better gene body coverage of scNOMeRe-seq data than smart-seq/smart-seq2, does it benefit this early embryo study? Could the authors please show some cases?
3. Line 84-88, the authors reported the DNA data of scNOMeRe-seq showed better genome coverages than those in scCOOL-seq, is there any normalization? There are several parameters impacting the genome coverage e.g., sequencing depth.
4. It is not surprising that PG blastomeres showed delayed development. It should be investigated further e.g., what the differences between PG embryos and normal embryos with integrative information e.g., specific region of Acc, Met and differential regulators.
5. About Acc clustering (Line 111-129, Extended Data Fig. 3), have the authors tried K-nearest neighbor (KNN) based clustering? The authors state that the cluster_2 (higher Acc level) probably linked to DNA duplication, can the authors exclude this results reflect an enzyme-, PCR- or sequencing-derived artefact? Besides the NDRs, it would be more convincing to check whether RNA and Met data in cluster_2 also showed evidences to support this point since this is an integrated data with multi-omics layers.
6. It is unclear to me why the Acc clustering was quite well-organized with the respective to the cell stage whereas Met and RNA clustering showed relatively mixed (Fig. 1d and Extended Data Fig. 2g).
7. How about the open chromatin accessible regions gained during development, and what about those lost ones? What is the correlation among these gained and lost Acc with Met and RNA data?
8. It is good to see the reconstruction of genetic lineages in mouse early embryos using single-cell Met datasets referred to a published method, however, besides the corrections and patterns, it would be also interesting to see the distinct molecular features (each layer) of daughter cells originated from different mother cell. Moreover, it would be more convincing to have more blastomeres to validate.
9. The statement "Moreover, the correlations between blastomeres from the same grandmother cells were higher than those of blastomeres from different grandmother cells in 8-cell embryos (Fig. 2h)." is not supported by Fig. 2h, at least as presented.
10. The statement in line 263 "Considering the global decreases in Acc during the ZGA process", are there any figures to support this? It has been reported distinct regulatory patterns during ZGA (DOI: 10.1038/s41467-018-08244-0), the authors should double check this statement.
11. It has been reported that many TFs important in mouse preimplantation embryos, for example, Hippo/Yap1, Nr5a2 and Rarg are important in the lineage segregation of the ICM and the TE in the mouse. The authors should investigate more putative TFs based on this multi-omics

dataset, and also further check whether their binding motifs enriched in any accessible regions which showed high correlation with other omics layers. Besides mouse, there are also many human early embryo studies, what kind of species differences in this important development process need further investigation based on this interesting dataset.

12. In Fig. 5i, several important TFs e.g., Klf5, Klf6 in group IV stated as showing no difference in activity but higher expression level in ICM cells, however, this statement was not supported by Fig. 5i, at least as presented. There is a typo in the y-axis label in Fig. 5i.

13. Will these triple layers of omics data help provide more accurate descriptions/definitions of "single-cell state"?

Reviewer #2 (Remarks to the Author):

The authors reported a single-cell multi-omics technology (scNOMeRe-seq) that enabled profiling of mouse preimplantation embryo cells for genome-wide chromatin accessibility, DNA methylation and RNA expression. They applied this new platform and analyzed the global dynamics of epigenetic molecular layers. The authors constructed a zygotic genome activation (ZGA)-associated regulatory network and revealed coordination among multiple epigenetic layers, transcription factors (TFs) and repeat elements that instruct the proper ZGA process. The analysis also revealed the partial ZGA and abnormal development of PG embryos, the parental specific allelic gene expression and epigenetic profiles, and enabled reconstruction of genetic lineages that reveals the source of heterogeneity in early embryos. Finally, they investigated the candidate TFs responsible for the establishment of differential regulatory networks in the ICM and the TE lineages.

In summary, this work is expected to further facilitate the single cell genomics and multi-omics research field and to improve the fundamental understanding of epigenetic regulation in early embryos.

Minor points:

1. Various single-cell epigenomic methods have been developed in the past several years to profile epigenetic molecular layers, providing opportunities to explore the associations among molecular regulatory layers.

What are the major advantages and disadvantages of scNOMeRe-seq compared with the author's previous technologies including the one mentioned in the manuscript – scCOOL-seq, and with other single-cell multi-omics sequencing technologies (eg. scTrio-seq, scNMT-seq, scNOMe-seq, etc.)?

2. Fig. 2h: The manuscript states that "the correlations in Met levels between blastomeres from the same mother cells were higher in 8-cell embryos than in 4-cell and late 4-cell embryos". The authors need to provide proper statistics of these analyses including p-value.

3. Fig. 3c /3h. More details about determining paternal alleles and maternal alleles are needed, assuming that these embryos are from several crosses.

4. Fig. 3h. The authors state that "the correlations between maternal Met and Expr at gene body regions were clearly weaker in nonmaternal genes than in maternal genes". The authors should provide p-value and proper statistics.

5. Fig. 4f. "Klf4, Nkx3-2, Nr5a2 and Rarg showed high TF activity and high expression levels in 2-cell embryos compared to zygotes". It seems that the expression of Klf4, Nkx3-2, Nr5a2 in 2C-cells is similar to in the zygotes. It may be better to use Z-score, instead of TPM.

6. Fig. 5c/d: "Notably, all of the known enhancers for three key ICM/TE TFs (Pou5f1, Nanog, and Cdx2) that we analyzed were revealed to be present in preimplantation embryos or in embryonic stem cells, confirming that the CREs identified by our correlation analysis could cover known active

enhancers ". The correlation coefficient(r) of the expression level of Pou5f1 and chromatin accessibility of positive-correlated CREs labelled in (c) is low(0.35)?

7. The authors discover that Klf4 could be a maternal factor and have important functions in ZGA. They can design experiments to verify it. Similarly, for the newly identified TFs that affect the ICM/TE separation, the author can also do experimental verification. These validation results will provide further supports to using the cNOMeRe-seq.

Reviewer #3 (Remarks to the Author):

This manuscript is entitled "Single-cell multiomics sequencing reveals the functional regulatory landscape of early embryos." The manuscript describes a series of experiments to reveal the profiles of genome-wide chromatin accessibility, DNA methylation and RNA expression in the same individual cells of the embryo through to blastocyst stage. Several comments are generated by review of the manuscript:

- 1) The manuscript would benefit from an introductory figure either in the main text or supplementary data that depicts the experimental design, embryo numbers and progression of experiments.
- 2) Embryo numbers for different experiments should be clearly indicated in both the experimental methods/design and the figure legends.
- 3) A figure depicting a model that derives from the data as a summary figure would enhance the manuscript greatly.

Overall, the manuscript is well written and the data are intriguing. It is notable, however, that the testing of the findings via use of inhibitors or loss-of-function or gain-of-function genetics has not been incorporated into the manuscript to provide causation proof. Thus, the manuscript largely correlates molecular changes with development and does not provide further substantiation. Nonetheless, the experiments are illuminating and provide a foundation of data for further exploration and generation of hypotheses.

The point-by-point responses:

We thank the reviewers for their comments. Below are our point-by-point responses. The reviewers' comments are in plain text and our responses are in blue colored. The cross references to the manuscript are bold and underlined. (Line numbers mentioned in the responses may not coincide with the original line numbers.)

Reviewers' comments:

Reviewer #1 (Remarks to the Author):

The present manuscript by Wang et al. developed a single-cell multiomics sequencing technology (scNOMeRe-seq) that enables profiling of genome-wide chromatin accessibility, DNA methylation and RNA expression in the same individual cell. By applying this method, the authors studied the global dynamics of different molecular layers and their associations in mouse preimplantation embryos. Based on a comprehensive amount of data and detailed analysis, this paper deciphers a single-cell triple-omics map of chromatin accessibility, DNA methylation and RNA expression during mouse preimplantation development and provides many specific and novel insights into the understanding of epigenetic regulation in early embryos. The dataset would be of interest to others in the community, whereas, its weak points are the lack of finding the specific TFs/CREs/Met and the largely confirmatory nature of the results. It will substantially improve the manuscript if more analysis results of the datasets and validation work can be presented. Below please find my specific points that I feel should be addressed prior to publication.

1. This manuscript reported the method "scNOMeRe-seq" which combined scNOMe-seq and MATQ-seq, the authors should demonstrate the strength of this method compared with other similar method e.g., scNMT-seq (doi: 10.1038/s41467-018-03149-4). Besides, more evaluation and comparison including chromatin accessibility and methylation should be presented to support this method a valid and reproducible approach. For example, methylation coverage, whether the accessibility data affected by endogenous GpC methylation, Met and Acc profiles at DNase hypersensitive sites etc.

Response:

We thank the Reviewer for the constructive suggestions. We have added more comparisons between our method with other similar methods to more clearly demonstrate the strength of our method (revised Extended Data Fig. 1). For the RNA data, we compared our method with Smart-seq2 (published single omics of mouse early embryos data) and scNMT-seq (employed Smart-seq2 to detect transcriptome of mouse embryonic stem cell data)(Clark et al., 2018, Nat Commun; Deng et al., 2014, Science). The results showed that our method could detect more genes with high accuracy and reproducibility and had better gene body coverage (Extended Data Fig. 1h). For the DNA data, our method together with published scNMT-seq and scCOOL-seq are all derived from NOMe-seq method, which used GpC methylase treatment combined with bisulfate sequencing to assess the DNA methylome (the methylation level of WCG sites) and genome wide chromatin accessibility (the methylation level of GCH sites)(Clark et al., 2018, Nat Commun; Guo et al., 2017, Cell Res; Kelly et al., 2012, Genome Res). The results showed that our method could simultaneously detect over 15% genomic WCG/GCH sites (WCG: 3.49 million sites, 15.8% of genomic coverage; GCH: 31.0 million sites, 15.5% of genomic coverage on average per cell) at around 3× sequencing depth with improved capture efficiency compared with scNMT-seq and scCOOL-seq (Extended Data Fig. 1e)(Clark et al., 2018, Nat Commun; Guo et al., 2017, Cell Res). In order to support this method a valid and reproducible approach, we previously showed that great fraction of NDRs identified in our method was overlapped with published defined open chromatin in early embryos at each corresponding stage (liDNase-seq, ATAC-seq and scCOOL-seq) (Extended Data Fig. 3f) (Guo et al., 2017, Cell Res; Lu et al., 2016, Cell; Wu et al., 2016, Nature). Following the Reviewer’s suggestion, we further calculated the chromatin accessibility and DNA methylation level around published DHSs/open regions (Lu et al., 2016, Cell; Wu et al., 2016, Nature). The results showed a high GCH and low WCG methylation level at the previously defined DHSs and open chromatin regions (Extended Data Fig. 1d), further supporting our method a valid and reproducible approach. We have modified our manuscript in Line 83-90 of revised manuscript, **“The DNA data showed that our method could simultaneously detect over 15% genomic WCG/GCH sites (WCG 3.49 million, 15.8%; GCH 31.0 million, 15.5% on average per cell at around 3× sequencing depth) with improved capture efficiency compared with single-cell nucleosome, methylation and transcription sequencing (scNMT-seq) and single-cell chromatin overall omic-scale landscape sequencing (scCOOL-seq) (Extended Data Fig. 1e)^{4,23}. Moreover, the DNA data”**

showed a high GCH and low WCG methylation level at the previously defined DNase hypersensitive sites and open chromatin, supporting our method a valid and reproducible approach (Extended Data Fig. 1d)^{6,10}.”

The assessment of chromatin accessibility with GCH methylation in NOME-seq has been fully validated and well accepted (Clark et al., 2018, Nat Commun; Guo et al., 2017, Cell Res; Kelly et al., 2012, Genome Res). To further evaluate whether the accessibility data affected by endogenous GpC methylation in this study, we have calculated the methylation level of GCH sites in the scBS-seq data (without GpC methylase treatment, obtained from FITC-microinjected embryos) and scNOMERe-seq data (with GpC methylase treatment) (Fig. R1). Clearly, the GCH methylation level of scBS-seq samples (consider as endogenous GCH level) is less than 1% and extremely lower than that of scNOMERe-seq samples (Fig. R1), supporting that the endogenous GCH methylation would not affect the chromatin accessibility assessment.

Figure R1. Box plot showing the methylation level of GCH sites in the scBS-seq data (without GpC methylase) and the DNA part of scNOMERe-seq data (with GpC methylase). Mean ± SD are shown.

2. Extended Data Fig. 1d showed better gene body coverage of scNOMERe-seq data than smart-seq/smarts-seq2, does it benefit this early embryo study? Could the authors please show some cases?

Response:

We thank the Reviewer for the comment. The gene body was more evenly detected by our method (used MATQ-seq for RNA profiling in scNOMERe-seq) than the Smart-seq2 method, consistent with the

reported MATQ-seq method (Sheng et al., 2017, Nature methods). In principle, a better gene body coverage could contribute a more accurate evaluation of the expression level (Hendrickson et al., 2017, Nat Genet). The accurate calculation of gene expression level is fundamentally important for this study, benefiting our exploration of the interplay between transcriptome with different epigenetic molecular layers. Moreover, the better gene body coverage may also contribute to detect more genes compared to other methods (as shown in our revised Extended Data Fig. 1h).

3. Line 84-88, the authors reported the DNA data of scNOMeRe-seq showed better genome coverages than those in scCOOL-seq, is there any normalization? There are several parameters impacting the genome coverage e.g., sequencing depth.

Response:

We thank the Reviewer for the suggestion. As suggested, we have added the results of sequencing depth normalized genome coverage of our DNA data in the revised Extended Data Fig. 1e (please also see our response to the Question 1). The results showed that our method could simultaneously detect over 15% genomic WCG/GCH sites (WCG: 3.49 million sites, 15.8% of genomic coverage; GCH: 31.0 million sites, 15.5% of genomic coverage on average per cell) at around 3× sequencing depth with improved capture efficiency compared with scNMT-seq and scCOOL-seq (Extended Data Fig. 1e)(Clark et al., 2018, Nat Commun; Guo et al., 2017, Cell Res).

4. It is not surprising that PG blastomeres showed delayed development. It should be investigated further e.g., what the differences between PG embryos and normal embryos with integrative information e.g., specific region of Acc, Met and differential regulators.

Response:

We thank the Reviewer for the comment. Following the Reviewer's suggestion, we have investigated the difference between PG embryos and normal embryos at different molecular layers as shown below. We found most of differentially expressed genes (DEGs, 93.71% of PG upregulated genes, 89.53% of PG downregulated genes, FDR<0.01, Fold change>=2) between PG embryos and normal embryos were transient altered at one stage of PG cells, and back to normal range in the next stage (Fig. R2a, R3a). Since both the PG embryos and normal embryos of the same stage were collected at the same time points, we suspected that the transient DEGs might reflect the nature of delayed development of the PG

embryos. Among those DEGs, 31 known paternal specific expressed genes were found in the PG downregulated genes, while two known maternal specific expressed genes were found in the PG upregulated genes (Fig. R2b, R3b). Next, we explored whether those DEGs were caused by the abnormal epigenetic reprogramming of PG embryos. We found relatively more PG-hypermethylated-TSSs and PG-hypomethylated-genebodies (FDR<0.01, Differential Met>10%) were constantly identified in PG downregulated genes compared to that in PG upregulated genes, indicating the difference of DNA methylation level could partially explain the difference of gene expression (Fig. R2c-f, R3c-f). However, we did not find clear relationships between the expression level and chromatin accessibility of its TSS region for these DEGs (Fig. R2g-h, R3g-h).

Figure R2: PG upregulated genes. Heat map of the RNA expression level of PG upregulated genes (a), and PG upregulated known imprinted genes (b); Dynamics of the DNA methylation level of PG upregulated genes at TSS (c and d) and gene body (e and f); g-h. Dynamics of the chromatin accessibility level of PG upregulated genes at TSS. DEG.type: stage specific DEGs

(2-cell, 4-/L4-cell, 8-cell), NSDEGs (non-stage-specific DEGs); PG_hyper: PG hypermethylated (Met: WCGs, Acc: GCHs); PG_hypo: PG hypomethylated (Met: WCGs, Acc: GCHs); no.diff: No significant difference; n.d.: not determined.

Figure R3: PG downregulated genes. Heat map of the RNA expression level of PG downregulated genes (a), and PG downregulated known imprinted genes (b); Dynamics of the DNA methylation level of PG downregulated genes at TSS (c and d) and gene body (e and f); g-h. Dynamics of the chromatin accessibility level of PG downregulated genes at TSS. DEG.type: stage specific DEGs (2-cell, 4-/L4-cell, 8-cell), NSDEGs (non-stage-specific DEGs); PG_hyper: PG hypermethylated (Met: WCGs, Acc: GCHs); PG_hypo: PG hypomethylated (Met: WCGs, Acc: GCHs); no.diff: No significant difference; n.d.: not determined.

5. About Acc clustering (Line 111-129, Extended Data Fig. 3), have the authors tried K-nearest neighbor (KNN) based clustering? The authors state that the cluster_2 (higher Acc level) probably linked to DNA duplication, can the authors exclude this results reflect an enzyme-, PCR- or sequencing-derived artefact? Besides the NDRs, it would be more convincing to check whether RNA

and Met data in cluster_2 also showed evidences to support this point since this is an integrated data with multi-omics layers.

Response:

We thank the Reviewer for the comment. The KNN is an alternative method for classification. However, it requires training dataset to learn the model before clustering. As we did not know the identity of each cell, we cannot construct the prediction model for the samples. Therefore, we use the K-means method to cluster the cells into two groups, which do not need prior identity of each cell. To determine the optimal number of clusters, we used Average Silhouette method as shown in Fig. R4. The results show two clusters (k=2) maximize the average silhouette values for most of stages (6 out of 8). For the rest two stages (Zygote and L4-cell), the average silhouette value of two clusters, as the second optimal number of clusters, are very close to that of the optimal number of clusters (k=3). In order to keep the consistency of the number of clusters, we used two clusters (k=2) for all stages we analyzed.

Figure R4: Average silhouette values of K-means clustering of Acc for each stage.

We thank the Reviewer for pointing out other possibilities of the cluster_2. Clearly, the level of Acc and Met from the same single cell were measured by the methylation level of GCH and WCG sites extracted from the same DNA sequencing data. Since the DNA methylome from the cluster_2 cells were similar to the cluster_1 cells of the same stage, we suspected that the higher level of Acc from cluster_2 was not likely caused by the PCR- or sequencing- derived artefacts, otherwise the Met data should be also found noticeable difference between cluster_1 cells and cluster_2 cells. In addition, we found most of cluster_2 cells were not come from some specific embryos, but mixed with cluster_1 cells in the same embryo (Fig. R5 Left). As the cells from the same embryo were treated with the same

batch of enzyme, as well as the same reaction conditions, indicating that the high level of Acc from cluster_2 might not reflect the enzyme derived artefacts, but genuine biological difference, which could be explained by the DNA duplication (Luo et al., 2017, Hum Mol Genet). To provide more cell cycle evidence, we inferred cell cycle of each cell with our RNA data (Fig. R5 Right). However, the results showed that the majority of cells from both clusters were classified into G2M phase, and no difference in the cell cycle results between cluster_1 and cluster_2 was noticed (Fig. R5 Right). To be more rigorous, we have rephrased this statement in Line 116-127 of the revised manuscript, “Notably, the cells from the cluster with the relatively higher Acc level (cluster 2) consistently showed lower correlations between Acc and Expr at the TSS regions, without differences in global Met levels, and correlations between Met and Expr were observed between the two clusters, suggesting that Acc changes in the cells of cluster 2 were irrelevant to the transcriptional regulation, which could be derived from biological differences, such as DNA duplication³⁰, or other unknown technical artefacts (Extended Data Fig. 3b). Furthermore, we detected the nucleosome-depleted regions (NDRs) using an aggregated Acc dataset from single cells in each cluster at each stage. Regardless of the genome coverage, cluster 1 (low Acc level and high correlation between Acc and Expr) exhibited more NDRs than cluster 2 for each stage (Extended Data Fig. 3c-d). The NDRs in cluster 1 at each stage showed greater fractions overlapping with previously defined open chromatin in early embryos than those in cluster 2 (Extended Data Fig. 3e-f)^{4,6,10}.”

Figure R5: Left. Number of embryos containing only cluster_1, cluster_2 cells, or mixed cluster_1 and cluster_2; Right. Cell cycles of cluster_1 and cluster_2 cells with *cyclone()* in Scran R package.

6. It is unclear to me why the Acc clustering was quite well-organized with the respective to the cell stage whereas Met and RNA clustering showed relatively mixed (Fig. 1d and Extended Data Fig. 2g).

Response:

We thank the Reviewer for the comment. Previously, we used cellular pairwise spearman correlation coefficients to perform PCA and unsupervised hierarchical clustering for each molecular layer. To improve the clustering appearance, we have tried to cluster RNA directly using the expression matrix of high variable genes. The results showed an improved RNA clustering with well separated stages and defined cell type, so we have updated these results in our revised Fig. 1d and the text of corresponding Method section in Line 859-862, **“PCA and hierarchical clustering were performed to analyze cell populations with the Expr data. Genes expressed in fewer than 6 cells were discarded. PCA was performed with the expression matrix of high variable genes (coefficient of variation ≥ 1) using the *pcaMethods*⁴⁰ R package. The *hclust* function with the *ward.D2* method was used for unsupervised hierarchical clustering.”**

As for the Met and Acc clustering, actually, previous Met clustering results showed major separation between blastocyst and pre-blastocyst (Zygote to Morula stages), and the Acc clustering showed a better separation pattern among different stages than the Met clustering, consistent with previous findings (Guo et al., 2017, Cell Res). To have a better presentation of Met clustering, we have treated the Met unsupervised hierarchical clustering result with *reorder.dendrogram()* function in R. The modified result has been updated in revised Fig. 1d.

7. How about the open chromatin accessible regions gained during development, and what about those lost ones? What is the correlation among these gained and lost Acc with Met and RNA data?

Response:

We thank the Reviewer for the question. We have analyzed the NDR changes between two consecutive stages (Fig. R6a). The results showed both Proximal_NDRs and Distal_NDRs were more dynamic than TSS_NDRs during development. And the number of gained NDRs was the highest in the transition from Zygote to 2-cell stage, accompanied with ZGA process.

We further investigated the changes of Acc, Met and Expr among these gained and lost NDRs. Clearly, the Acc level of gained NDRs was increased in the latter stage of two consecutive stages, contrast to the decreased Acc level of the lost NDRs in the latter stage (Fig. R6b). The dynamic of NDRs were generally associated with the changes of Met level and gene expression. The gained NDRs showed

lower Met level and higher expression level in the latter stage, while the lost NDRs showed higher Met level and lower expression level in the latter stage (Fig. R6 c-d).

Figure R6: (a) Bar plot showing the number of gained, lost and maintained NDRs between two consecutive development stages. Gained (or Lost) NDRs: the NDRs only found in the latter (or earlier) stage of two consecutive stages; Maintained NDRs: the overlapped NDRs between two consecutive development stages. Boxplot showing the differential Acc level (b), Met level (c) and Expr (d) of gained and lost NDRs between two consecutive development stages. y.axis, latter stage minus earlier stage of two consecutive development stages; P, p value, Student t-test.

8. It is good to see the reconstruction of genetic lineages in mouse early embryos using single-cell Met datasets referred to a published method, however, besides the corrections and patterns, it would be also interesting to see the distinct molecular features (each layer) of daughter cells originated from different mother cell. Moreover, it would be more convincing to have more blastomeres to validate.

Response:

We thank the Reviewer for raising this interesting question. Following the Reviewer's suggestion, we have performed comparisons to explore the distinct molecular features of daughter cells originated from different mother cells. Specifically, we first identified the differentially expressed genes (FDR<5%, FC>=2), differentially methylated TSSs (FDR<1%, Difference>10%) and differentially accessible TSSs (FDR<1%, Difference>10%) between the daughter cells (for 4-cell and L4-cell stages)/granddaughter cells (for 8-cell stage) from different 2-cell mother cells for each embryo, then we overlapped the DEGs/differential Met/Acc TSSs from different embryos of the same stage to test which gene/TSS was constantly biased expressed/epigenetic modified. However, no genes were biased expressed in all embryos of the same stage. Only few genes were constantly biased expressed in over half of embryos (5 genes in 4-cell stage, 4 genes in L4-cell stage and 7 genes in 8-cell stage), and no overlapped genes were found among the three stages. Similar results were found for the differential Met TSSs and Acc TSSs. Although some TSSs showed biased modified in all embryos of the same stage (such as the Diff Met TSSs of 8-cell stage and the Diff Acc TSSs of L4-cell stage), no more than three embryos of those stages were analyzed. Moreover, there was also no overlapping between DEGs and differentially modified TSSs. Together, our preliminary results suggested that there might be no fixed molecular features between the daughter cells originated from different mother cells. However, to fully address this interesting question would require further investigation.

Figure R7: Bar plot showing the number of differentially expressed genes (upper), DNA methylated TSSs (middle) and accessible TSSs (bottom). Venn diagram showing the number of overlapped genes (upper) and TSSs (middle and bottom) biased expressed/modified in over half of embryos of the same stage. For DEGs, $FDR < 5\%$, $FC \geq 2$; Diff Met/Acc TSSs, $FDR < 1\%$, $Difference > 10\%$.

As for the lineage reconstruction with single-cell DNA methylation datasets, the consistent correlation pattern was observed for all embryos we analyzed (for 4-/L4-cell embryo, $N=17$, including two PG embryos; for 8-cell embryo, $N=4$, including one PG embryo) without any exemption (Fig. R8). Further, we also used DNA methylation data of three 4-cell embryos and two 8-cell embryos with FITC microinjection to determine the correlation pattern of cells from the same 2-cell mother and the pattern of cells from the same 4-cell mother, validating the accurate use of single-cell DNA methylation datasets to reconstruct lineages of early embryos.

Figure R8: Heat map showing the Pearson correlation coefficients of z-scored DNA methylation level among cells in each embryo of 4-cell, L4-cell and 8-cell stages. Red labeled PG embryos; green labeled cells divided from the same 2-cell blastomere (FITC_4-cell_1-3# and FITC_8-cell_1#) or from the same 4-cell blastomere (FITC_8-cell_2#). The 4-cell_5# embryo was not shown here due to the insufficient cells for this embryo to perform this analysis.

In this study, we have showed that the correlations of the cells from the same mother inferred from Met data is higher than that of the cells from different mother at the transcriptome level, and the correlations of cells from the same grandmother is higher than that of cells from different grandmother (Fig. 2h). Further, we explored whether this correlation differences of transcriptome were enabled us to reconstruct genetic lineages. Interestingly, we found the correlation differences were highly conserved for each embryo (as shown in our revised new Extended Data Fig. 4). Specifically, the cells from the same mother cell are clustered closer than the cells from different mother cell, similarly, the cells from the same grandmother cell are clustered closer than the cells from different grandmother cell. These RNA clustering results are consistent with the tracing results by Met datasets. Therefore, these results demonstrated that both single-cell transcriptome and methylome datasets could be used to reconstruct

genetic lineages of early embryos. We have added this new finding in the revised manuscript in Line 185-188, **“Interestingly, we found the gradually increased transcriptome heterogeneity during the first three cleavages were highly conserved for each embryo, which enabled us to reconstruct genetic lineages of early embryos with single-cell transcriptome datasets (Extended Data Fig. 4).”**

9. The statement “Moreover, the correlations between blastomeres from the same grandmother cells were higher than those of blastomeres from different grandmother cells in 8-cell embryos (Fig. 2h).” is not supported by Fig. 2h, at least as presented.

Response:

We thank the Reviewer for the question. We are sorry for the confusion. For this statement, we were trying to point out the correlations between blastomeres from the same grandmother cells were higher than those of blastomeres from different grandmother cells in 8-cell embryos at the transcriptome level (average correlation coefficient of 0.8765 in same_grand and 0.8699 in diff_grand; same_grand vs diff_grand, $P=2e-07$), as shown in Fig. R9. We have modified this statement in the revised manuscript, Line 181-183, **“Moreover, the correlations between blastomeres from the same grandmother cells were higher than those of blastomeres from different grandmother cells in 8-cell embryos at the transcriptome level (Fig. 2h) (Student t-test, $P = 2e-07$).”**

Figure R9: Box plot showing the pairwise Spearman correlation coefficients of RNA expression level in 8-cell embryos. P, p value, Student t-test.

10. The statement in line 263 “Considering the global decreases in Acc during the ZGA process”, are there any figures to support this? It has been reported distinct regulatory patterns during ZGA (DOI: 10.1038/s41467-018-08244-0), the authors should double check this statement.

Response:

We thank the Reviewer for the comment. The global Acc level of embryos reached the lowest at the 2-cell stage when the major ZGA occurred, supported by Extended Data Fig. 2e-f. In addition, these results were consistent with previous findings used scCOOL-seq in the mouse preimplantation embryos (Guo et al., 2017, Cell Res).

Although the global decreases in Acc during the ZGA process, the Acc level of thousands of ZGA associated potential functional CREs (positively correlated CREs) was specifically increased in 2-cell embryos (Fig. 4a and Extended Data Fig. 6d). The distinct regulatory patterns during ZGA (reported in the Reviewer mentioned paper, DOI: 10.1038/s41467-018-08244-0), were identified by calculating the average chromatin accessibility level of the distinct expression patterns of human ZGA genes. In their results, the activation of major ZGA genes was overall associated with increased chromatin accessibility of promoters and enhancers, which is consistent with our findings.

To be clear, this statement “Considering the global decreases in Acc during the ZGA process and the characteristics of the negatively correlated CREs described above, these negative correlations seemed to simply reflect global changes in Acc rather than representing repressive regulation during ZGA.” was just our speculation about the negatively correlated CREs identified for ZGA process. We have removed this statement in our revised manuscript to avoid any confusion.

11. It has been reported that many TFs important in mouse preimplantation embryos, for example, Hippo/Yap1, Nr5a2 and Rarg are important in the lineage segregation of the ICM and the TE in the mouse. The authors should investigate more putative TFs based on this multi-omics dataset, and also further check whether their binding motifs enriched in any accessible regions which showed high correlation with other omics layers. Besides mouse, there are also many human early embryo studies, what kind of species differences in this important development process need further investigation based on this interesting dataset.

Response:

We thank the Reviewer for the suggestion. The putative TFs mentioned in this part were actually unbiased identified based on our multi-omics dataset. Specifically, to identify what TFs might play

important roles in the ICM and the TE lineage segregation, we first took advantage of our single cell multi-omics dataset containing both the gene expression and chromatin accessibility data of the same single cell and extracted the significantly correlated NDRs as potential functional CREs by calculating the correlations between the chromatin accessibility of NDRs and the expression level of corresponding ICM/TE differential expressed genes across cells during preimplantation development. Then, we performed motif enrichment analysis using these CREs with Homer2 software. Through our analysis, we have successfully identified total of 33 putative TFs, which might play a role in segregating the ICM and the TE lineages. Among them, many known TFs important for the lineage segregation have been revealed by our analysis, such as Gata, Klf, and Tead families, as well as Nr5a2, Rarg, shown in Fig. 5g.

For the TFs mentioned by the Reviewer, our result showed that Nr5a2 and Rarg were enriched in both ICM and TE distal CREs, but these two TFs were more enriched in ICM promoter CREs (Nr5a2, $P=1e-11$; Rarg, $P=1e-12$) than TE promoter CREs (Nr5a2, $P=1e-4$; Rarg, $P=1e-5$), suggesting their preferential role in activating ICM program. Hippo/Yap1 signal pathway has been shown in establishing TE lineage through regulating Tead4 activity (Nishioka et al., 2009, Developmental cell). In our results, we found that Tead4 was specifically enriched in TE CREs at both promoter ($P=1e-19$) and distal regions ($P=1e-28$) but ICM CREs, supporting that Tead4 played an important role in TE lineage establishment.

The dynamics of different molecular layers of human early embryos have been reported in recent years. Following the Reviewer's suggestion, we have tested whether mouse ICM and TE CREs would be open during human embryo development. First, we converted our mouse ICM/TE-CREs into human genome. We found over 80% promoter CREs were successfully converted into human genome, while around half distal CREs were converted into human genome (Fig. R10 a). Next, we overlapped the successfully converted CREs with human open regions (Wu et al., 2018, Nature). The results showed that mouse ICM/TE-CREs were significantly accessible in the human embryos (Fig. R10 b). Further, we performed motif enrichment analysis using the CREs open in human embryos and the rest converted CREs (Fig. R10 c-d). We found TFs such as CTCT, GATAs, KLFs and TEADs were enriched in the CREs open in human embryos, suggesting their conservative roles in the both human and mouse species (Fig. R10

d). We found ESRRRA and CRX are exclusively enriched in mouse CREs not open in human embryos, indicating these TFs might play species-specific role between mouse and human (Fig. R10 d). However, there is no available single cell multi-omics data simultaneously containing the transcriptome and chromatin accessibility from the same single cells of human early embryos, limiting our further comparisons in this study.

Figure R10: a-c. Bar plot showing the number of CREs of indicated type; d. TF enrichment analysis of indicated CREs.

12. In Fig. 5i, several important TFs e.g., Klf5, Klf6 in group IV stated as showing no difference in activity but higher expression level in ICM cells, however, this statement was not supported by Fig. 5i, at least as presented. There is a typo in the y-axis label in Fig. 5i.

Response:

We thank the Reviewer for the carefully reviewing. *Klf5/6* were higher expressed in TE cells. We have corrected this mistake as shown in our revised Fig. 5i and corresponding figure legend. And we also corrected the typo error of y-axis label in Fig. 5i.

13. Will these triple layers of omics data help provide more accurate descriptions/definitions of "single-cell state"?

Response:

We thank the Reviewer for raising this interesting question. We think that with the help of these triple layers of omics data of the same single cell would provide more accurate descriptions/definitions of single-cell state and beyond. As shown in our study, we used RNA reads and DNA reads, as well as SNP information, to infer the abnormal cells, and successfully identified aneuploid and parthenogenetic cells, providing the opportunity to study the consequence of these genomic abnormal cells at different molecular levels. In addition, we used the endogenous DNA methylation data to construct genetic lineage to know the history of the single cell in the 4-cell and 8-cell embryos; with these cell lineage information, we further revealed that the asymmetric cleavage was the potential major driver of the gradual increases in transcriptome heterogeneity among blastomeres that occur during the first three cleavages, and the Met maintenance was increased during DNA duplication at the 4-cell stage. Moreover, with the knowledge of the transcriptome and different layers of epigenome from the same single cells, we not only found the relationships among transcription, chromatin accessibility and DNA methylation at single cell level and at different allele level, but also successfully identified thousands of potential functional CREs and dozens of putative TFs during ZGA and cell lineage separation processes, enhancing the fundamental understanding of epigenetic regulation in early embryos.

Reviewer #2 (Remarks to the Author):

The authors reported a single-cell multi-omics technology (scNOMeRe-seq) that enabled profiling of mouse preimplantation embryo cells for genome-wide chromatin accessibility, DNA methylation and RNA expression. They applied this new platform and analyzed the global dynamics of epigenetic molecular layers. The authors constructed a zygotic genome activation (ZGA)-associated regulatory network and revealed coordination among multiple epigenetic layers, transcription factors (TFs) and repeat elements that instruct the proper ZGA process. The analysis also revealed the partial ZGA and abnormal development of PG embryos, the parental specific allelic gene expression and epigenetic profiles, and enabled reconstruction of genetic lineages that reveals the source of heterogeneity in early embryos. Finally, they investigated the candidate TFs responsible for the

establishment of differential regulatory networks in the ICM and the TE lineages.

In summary, this work is expected to further facilitate the single cell genomics and multi-omics research field and to improve the fundamental understanding of epigenetic regulation in early embryos.

Minor points:

1. Various single-cell epigenomic methods have been developed in the past several years to profile epigenetic molecular layers, providing opportunities to explore the associations among molecular regulatory layers.

What are the major advantages and disadvantages of scNOMeRe-seq compared with the author's previous technologies including the one mentioned in the manuscript – scCOOL-seq, and with other single-cell multi-omics sequencing technologies (eg. scTrio-seq, scNMT-seq, scNOMe-seq, etc.)?

Response:

We thank the Reviewer for the comments. Our method could parallel detect DNA methylation, chromatin accessibility and transcriptome from the same single cell. Compared with the most of published single-cell multi-omics sequencing technologies, our method could detect more layers of molecular with high quality and reproducibility. For instance, scCOOL-seq and scNOMe-seq are only able to detect DNA methylation and chromatin accessibility from the same single cell, and scTrio-seq is only able to detect DNA methylation (more focused on CpG enriched regions) and transcriptome simultaneously (Guo et al., 2017, Cell Res; Hou et al., 2016, Cell Res; Kelly et al., 2012, Genome Res). scNMT-seq is able to detect the same molecular layers with our method (Clark et al., 2018, Nat Commun). Despite the different orders of GpC treatment and RNA capture, the most important difference between our method and scNMT-seq is that different methods for RNA library preparation were employed, as we used MATQ-seq and scNMT-seq used SMART-seq2 (Picelli et al., 2013, Nature methods; Sheng et al., 2017, Nature methods).

In order to show the power of our method, we have compared the quality of our data of different molecular layers with corresponding dataset obtained from published scCOOL-seq, SMART-seq2 and scNMT-seq (revised Extended Data Fig. 1, please also see our response to the Reviewer 1#, question 1) (Clark et al., 2018, Nat Commun; Deng et al., 2014, Science; Guo et al., 2017, Cell Res). For the RNA data, our method could detect more genes with high accuracy and reproducibility and had better gene body coverage. For the DNA data, our method could detect over 15% genomic WCG/GCH sites (WCG: 3.49 million sites, 15.8% of genomic coverage; GCH: 31.0 million sites, 15.5% of genomic coverage on average per cell) at around 3× sequencing depth (8.74 Gb in average) with better capture efficiency compared with scNMT-seq and scCOOL-seq.

However, our method requires physically separating cytoplasm and nuclei at single cell level, which would increase the risk of loss sample and the chance of contamination.

2. Fig. 2h: The manuscript states that "the correlations in Met levels between blastomeres from the same mother cells were higher in 8-cell embryos than in 4-cell and late 4-cell embryos". The authors need to provide proper statistics of these analyses including p-value.

Response:

We thank the Reviewer for the comment. We have performed statistical analyses for the correlations in DNA methylation level between blastomeres. We found the correlations from the same mother cells in 8-cell embryos were higher than that in 4-/late 4-cell embryos (average correlation coefficient of 0.51 in 4-cell, n=20 pairs; 0.51 in late 4-cell, n=9 pairs; and 0.53 in 8-cell, n=12 pairs), but no statistical significance were observed when we used 5 kb bins to estimate the DNA methylation level (Fig. R11). Further, we titrated several different lengths of bins and calculated pairwise correlations. We found the pairwise correlation in 8-cell embryos is consistently higher than that in 4-/L4-cell embryos with different length of tiles (Fig. R11). Moreover, we found the difference of pairwise correlations between 8-cell embryos and 4-/L4-cell embryos was increased and showed more statistical significance with longer tiles. To support this statement, we have updated the pairwise correlation results of DNA methylation with 200 Kb bin in our revised Fig. 2h and corresponding text in the manuscript, Line 189-192, **“we notably observed that the correlations in Met levels between blastomeres from the same mother cells were higher in 8-cell embryos than in 4-cell and late 4-cell embryos (average**

correlation coefficient of 0.59 in 4-cell, 0.58 in late 4-cell and 0.74 in 8-cell; Student t-test, 4-cell vs 8-cell, P = 0.018; late 4-cell vs 8-cell, P = 0.013”.

Figure R11: Box plot showing the pairwise Spearman correlation coefficients of DNA methylation level of different lengths of bins from same mother cell. P, p value, Student t-test, compared with 8-cell stage.

3. Fig. 3c /3h. More details about determining paternal alleles and maternal alleles are needed, assuming that these embryos are from several crosses.

Response:

We thank the Reviewer for the comment. Following the Reviewer’s suggestion, we have added more details about determining parental alleles in the revised Method section, Line 962-978, “The embryos in this study are from 129S1 (paternal) mice × B6D2F1/J (F1 of C57BL6NJ × DBA2J, maternal) mice. Thus, these embryos should have backgrounds of 129S1 with mixed C57BL6NJ and DBA2J. The pipeline used to determine the parental origin assignment of sequencing data from the hybrid embryos was constructed as reported, which based on traceable hybrid SNP information⁵². Specifically, we downloaded the SNPs of 129S1 (paternal in this study), C57BL6NJ and DBA2J (C57BL6NJ × DBA2J, maternal in this study) from the website of Mouse Genome Project (ftp://ftp-mouse.sanger.ac.uk/REL-1211-SNPs Indels/). Only the informative SNPs could distinguish the paternal (129S1) and maternal (C57BL6NJ × DBA2J) genome (homozygous in parental alleles and paternal is different with maternal) were used in our analysis. For each mapped read covered the informative SNP site, the read was parsed according to the specific base at the SNP position, if the base matched the paternal allele, the read was assigned to paternal origin; if the base matched the maternal allele, the read was assigned to maternal origin. For

RNA-seq data, 172,319 SNPs within the exon regions were used to split the RNA mapped reads. The splitted allelic reads were further used to calculate the paternal and maternal expression level of each gene. For DNA data, 896,161 SNPs (SNP sites with C or T were discarded) in the whole genome were used to split the mapped reads to paternal and maternal origin. The splitted reads were further processed to calculate the paternal and maternal DNA methylation and chromatin accessibility level.”

4. Fig. 3h. The authors state that "the correlations between maternal Met and Expr at gene body regions were clearly weaker in nonmaternal genes than in maternal genes". The authors should provide p-value and proper statistics.

Response:

We thank the Reviewer for the comment. We have added box plots to show the correlation coefficients of allelic gene body Met vs Expr and performed statistical analyses between nonmaternal genes and maternal genes in the revised Extended Data Fig. 5g and corresponding figure legend.

5. Fig. 4f. "Klf4, Nkx3-2, Nr5a2 and Rarg showed high TF activity and high expression levels in 2-cell embryos compared to zygotes". It seems that the expression of Klf4, Nkx3-2, Nr5a2 in 2C-cells is similar to in the zygotes. It may be better to use Z-score, instead of TPM.

Response:

We thank the Reviewer for the suggestion. We have updated heat map of the TF expression with Z-score treatment in the modified Fig. 4f.

6. Fig. 5c/d: "Notably, all of the known enhancers for three key ICM/TE TFs (Pou5f1, Nanog, and Cdx2) that we analyzed were revealed to be present in preimplantation embryos or in embryonic stem cells, confirming that the CREs identified by our correlation analysis could cover known active enhancers ". The correlation coefficient(r) of the expression level of Pou5f1 and chromatin accessibility of positive-correlated CREs labelled in (c) is low (0.35)?

Response:

We thank the Reviewer for the comment. To compute the correlation coefficient (r) of the Acc level of each NDR and the expression of its corresponding gene, we adopted a previously published method in

(Argelaguet et al., 2019, Nature; Clark et al., 2018, Nat Commun) (please also see our Method section, Line 951-960). The NDRs with significant associations (FDR < 10%) were kept for downstream analysis.

The detailed information of positive-correlated CREs labelled in Fig. 5c has been collected from Supplementary Table 5 as shown below. All of the four CREs showed statistically significant correlations between the chromatin accessibility and *Pou5f1* expression. We agree that the correlation coefficients of these CREs are relatively low (the range of r is between 0.3 to 0.5). However, it seems common to observe relatively low correlations with single cells datasets. For example, the reported correlation coefficients of Acc vs Expr are between 0.2 to 0.5 in a previous report (Argelaguet et al., 2019, Nature). We think the relatively low correlation coefficient might be caused by the sparsity nature of single cell epigenetic omics, the unavoidable dropout from single cell RNA-seq, and the multi-layers of transcriptional regulation.

Table R1: The detailed information of *Pou5f1* related positive-correlated CREs

chr	start	end	symbol	r (Acc vs Expr)	P value (Acc vs Expr)	FDR (Acc vs Expr)	Correlation type	CRE type	No.
chr17	35569861	35570060	Pou5f1	0.35051163	0.000937723	0.020617572	pos.sig	ICM.CRE	#1
chr17	35634581	35634720	Pou5f1	0.329634608	0.006449953	0.064690553	pos.sig	ICM.CRE	#2
chr17	35639761	35640020	Pou5f1	0.394067855	0.002208283	0.034496974	pos.sig	ICM.CRE	#3
chr17	35640761	35641200	Pou5f1	0.4825338	1.42E-07	7.17E-05	pos.sig	ICM.CRE	#4

7. The authors discover that Klf4 could be a maternal factor and have important functions in ZGA. They can design experiments to verify it. Similarly, for the newly identified TFs that affect the ICM/TE separation, the author can also do experimental verification. These validation results will provide further supports to using the cNOMeRe-seq.

Response:

We thank the Reviewer for the suggestion. In this study, we have identified dozens of TFs potentially driving ZGA or ICM/TE separation. Notably, among of those TFs, many of them have been proved to be very important in those processes in previous studies, such as the Tead, Gata family, and Tcfap2c for driving TE lineage; Esrrb and Klf3/4 for driving ICM lineage, supporting the faithful and informative of our findings.

Klf4 is well known as a core transcription factor for pluripotency. A recent paper showed that Klf4 mutant embryos failed to form blastocyst with substantially reduced all cell lineages of epiblast, primitive endoderm and trophoctoderm, suggesting that Klf4 might play an important role in the very early stage of embryos (Ye et al., 2018, Nat Commun). Although there is no study claimed that Klf4 regulates ZGA process, but we found many Klf4 binding sites (published Klf4 ChIP-seq data of mouse embryonic stem cells) (Di Giammartino et al., 2019, Nature cell biology) were open in the 2-cell embryos (Fig. R12 Left), and Klf4 bound to the promoters of 317 ZGA genes (hypergeometric test, $p= 8.475122e-19$) (Fig. R12 Right), indicating that Klf4 may have important functions in ZGA.

Figure R12 Left: Heat map showing the number of Klf4 ChIP-seq peaks overlap with NDRs. Right: Venn diagram showing the overlaps between Klf4 ChIP-seq promoter peaks bound genes and ZGA genes.

We agree that functional validations would further provide more evidences to support the importance of our newly identified TFs in early embryo. However, it would be time consuming to perform experimental verification for those TFs, especially for some of them required conditionally knock-out experiments. We appreciate the Reviewer for this constructive suggestion and would like to test those newly identified TFs in our future study.

Reviewer #3 (Remarks to the Author):

This manuscript is entitled "Single-cell multiomics sequencing reveals the functional regulatory landscape of early embryos." The manuscript describes a series of experiments to reveal the profiles of genome-wide chromatin accessibility, DNA methylation and RNA expression in the

same individual cells of the embryo through to blastocyst stage. Several comments are generated by review of the manuscript:

1) The manuscript would benefit from an introductory figure either in the main text or supplementary data that depicts the experimental design, embryo numbers and progression of experiments.

Response:

We thank the Reviewer for the suggestion. We have added an introductory figure in the modified Fig. 1a to show the experimental design, embryo numbers and progression of experiments.

2) Embryo numbers for different experiments should be clearly indicated in both the experimental methods/design and the figure legends.

Response:

We thank the Reviewer for the suggestion. We have added the embryo numbers of different experiments in the experimental methods/design and the figure legends as suggested.

3) A figure depicting a model that derives from the data as a summary figure would enhance the manuscript greatly.

Response:

We thank the Reviewer for the suggestion. We have added our proposed models for regulating the processes of ZGA and ICM/TE separation in our modified Figure 4 and 5, respectively.

Overall, the manuscript is well written and the data are intriguing. It is notable, however, that the testing of the findings via use of inhibitors or loss-of-function or gain-of-function genetics has not been incorporated into the manuscript to provide causation proof. Thus, the manuscript largely correlates molecular changes with development and does not provide further substantiation. Nonetheless, the experiments are illuminating and provide a foundation of data for further exploration and generation of hypotheses.

References:

- Argelaguet, R., Clark, S.J., Mohammed, H., Stapel, L.C., Krueger, C., Kapourani, C.A., Imaz-Rosshandler, I., Lohoff, T., Xiang, Y., Hanna, C.W., *et al.* (2019). Multi-omics profiling of mouse gastrulation at single-cell resolution. *Nature* *576*, 487-491.
- Clark, S.J., Argelaguet, R., Kapourani, C.A., Stubbs, T.M., Lee, H.J., Alda-Catalinas, C., Krueger, F., Sanguinetti, G., Kelsey, G., Marioni, J.C., *et al.* (2018). scNMT-seq enables joint profiling of chromatin accessibility DNA methylation and transcription in single cells. *Nat Commun* *9*, 781.
- Deng, Q., Ramskold, D., Reinius, B., and Sandberg, R. (2014). Single-cell RNA-seq reveals dynamic, random monoallelic gene expression in mammalian cells. *Science* *343*, 193-196.
- Di Giammartino, D.C., Kloetgen, A., Polyzos, A., Liu, Y., Kim, D., Murphy, D., Abuhashem, A., Cavaliere, P., Aronson, B., Shah, V., *et al.* (2019). KLF4 is involved in the organization and regulation of pluripotency-associated three-dimensional enhancer networks. *Nature cell biology* *21*, 1179-1190.
- Guo, F., Li, L., Li, J., Wu, X., Hu, B., Zhu, P., Wen, L., and Tang, F. (2017). Single-cell multi-omics sequencing of mouse early embryos and embryonic stem cells. *Cell Res* *27*, 967-988.
- Hendrickson, P.G., Dorais, J.A., Grow, E.J., Whiddon, J.L., Lim, J.W., Wike, C.L., Weaver, B.D., Pflueger, C., Emery, B.R., Wilcox, A.L., *et al.* (2017). Conserved roles of mouse DUX and human DUX4 in activating cleavage-stage genes and MERVL/HERVL retrotransposons. *Nat Genet* *49*, 925-934.
- Hou, Y., Guo, H., Cao, C., Li, X., Hu, B., Zhu, P., Wu, X., Wen, L., Tang, F., Huang, Y., *et al.* (2016). Single-cell triple omics sequencing reveals genetic, epigenetic, and transcriptomic heterogeneity in hepatocellular carcinomas. *Cell Res* *26*, 304-319.
- Kelly, T.K., Liu, Y., Lay, F.D., Liang, G., Berman, B.P., and Jones, P.A. (2012). Genome-wide mapping of nucleosome positioning and DNA methylation within individual DNA molecules. *Genome Res* *22*, 2497-2506.
- Lu, F., Liu, Y., Inoue, A., Suzuki, T., Zhao, K., and Zhang, Y. (2016). Establishing Chromatin Regulatory Landscape during Mouse Preimplantation Development. *Cell* *165*, 1375-1388.
- Luo, H., Xi, Y., Li, W., Li, J., Li, Y., Dong, S., Peng, L., Liu, Y., and Yu, W. (2017). Cell identity bookmarking through heterogeneous chromatin landscape maintenance during the cell cycle. *Hum Mol Genet* *26*, 4231-4243.
- Nishioka, N., Inoue, K., Adachi, K., Kiyonari, H., Ota, M., Ralston, A., Yabuta, N., Hirahara, S., Stephenson, R.O., Ogonuki, N., *et al.* (2009). The Hippo signaling pathway components Lats and Yap pattern Tead4 activity to distinguish mouse trophectoderm from inner cell mass. *Developmental cell* *16*, 398-410.
- Picelli, S., Bjorklund, A.K., Faridani, O.R., Sagasser, S., Winberg, G., and Sandberg, R. (2013). Smart-seq2 for sensitive full-length transcriptome profiling in single cells. *Nature methods* *10*, 1096-1098.
- Sheng, K., Cao, W., Niu, Y., Deng, Q., and Zong, C. (2017). Effective detection of variation in single-cell transcriptomes using MATQ-seq. *Nature methods* *14*, 267-270.
- Wu, J., Huang, B., Chen, H., Yin, Q., Liu, Y., Xiang, Y., Zhang, B., Liu, B., Wang, Q., Xia, W., *et al.* (2016). The landscape of accessible chromatin in mammalian preimplantation embryos. *Nature* *534*, 652-657.
- Wu, J., Xu, J., Liu, B., Yao, G., Wang, P., Lin, Z., Huang, B., Wang, X., Li, T., Shi, S., *et al.* (2018). Chromatin analysis in human early development reveals epigenetic transition during ZGA. *Nature* *557*, 256-260.
- Ye, B., Liu, B., Hao, L., Zhu, X., Yang, L., Wang, S., Xia, P., Du, Y., Meng, S., Huang, G., *et al.* (2018). Klf4 glutamylation is required for cell reprogramming and early embryonic development in mice. *Nat Commun* *9*, 1261.

REVIEWERS' COMMENTS

Reviewer #1 (Remarks to the Author):

The authors addressed all of my comments in a satisfying manner, and I recommend the revised manuscript for publication. I wish the authors luck with this interesting paper and their follow up studies.

Reviewer #2 (Remarks to the Author):

This manuscript extensively analyzed the single cell data that they generated from mouse preimplantation embryos. The major conclusions are sound.

I do not have further comments on the manuscript.

Pentao Liu

Reviewer #3 (Remarks to the Author):

The authors have responded to each of the comments and suggestions from this reviewer.

REVIEWERS' COMMENTS

Reviewer #1 (Remarks to the Author):

The authors addressed all of my comments in a satisfying manner, and I recommend the revised manuscript for publication. I wish the authors luck with this interesting paper and their follow up studies.

Response:

We thank the Reviewer for the positive comments.

Reviewer #2 (Remarks to the Author):

This manuscript extensively analyzed the single cell data that they generated from mouse preimplantation embryos. The major conclusions are sound.

I do not have further comments on the manuscript.

Pentao Liu

Response:

We thank the Reviewer for the positive comments.

Reviewer #3 (Remarks to the Author):

The authors have responded to each of the comments and suggestions from this reviewer.

Response:

We thank the Reviewer for the positive comments.